# Kinase inhibition profiles as a tool to identify kinases for specific phosphorylation sites

Nikolaus A. Watson [1], Tyrell N. Cartwright [1], Conor Lawless [2], Marcos Cámara-Donoso[1], Onur Sen [1], Kosuke Sako [3], Toru Hirota[3], Hiroshi Kimura [4] & Jonathan M.G. Higgins [1✉]

There are thousands of known cellular phosphorylation sites, but the paucity of ways to identify kinases for particular phosphorylation events remains a major roadblock for understanding kinase signaling. To address this, we here develop a generally applicable method that exploits the large number of kinase inhibitors that have been profiled on near-kinome-wide panels of protein kinases. The inhibition profile for each kinase provides a fingerprint that allows identification of unknown kinases acting on target phosphosites in cell extracts. We validate the method on diverse known kinase-phosphosite pairs, including histone kinases, EGFR autophosphorylation, and Integrin β1 phosphorylation by Src-family kinases. We also use our approach to identify the previously unknown kinases responsible for phosphorylation of INCENP at a site within a commonly phosphorylated motif in mitosis (a non-canonical target of Cyclin B-Cdk1), and of BCL9L at S915 (PKA). We show that the method has clear advantages over in silico and genetic screening.

[1] Biosciences Institute, Faculty of Medical Sciences, Newcastle University, Newcastle upon Tyne NE2 4HH, UK. [2] Wellcome Centre for Mitochondrial Research, Faculty of Medical Sciences, Newcastle University, Newcastle upon Tyne NE2 4HH, UK. [3] The Cancer Institute, Japanese Foundation for Cancer Research, Koto, Tokyo 135-8550, Japan. [4] Cell Biology Center, Institute of Innovative Research, Tokyo Institute of Technology, Yokohama, Kanagawa 226-8503, Japan. ✉email: jonathan.higgins@newcastle.ac.uk

Protein phosphorylation is a vital regulatory system in cells, with 90% of all proteins undergoing phosphorylation[1]. Advances in phosphoproteomics mean that tens of thousands of phosphorylation sites are now known[2,3]. Large catalogues of specific phosphorylation events that occur within particular cell types or organelles, or during specific treatments or phases of the cell cycle, have been generated[4–8]. However, understanding the biology controlled by a particular phosphorylation event depends on being able to identify the kinase(s) responsible for catalyzing it. Unfortunately, the methods available for assigning kinase-phosphosite dependencies are limited. Consequently, our ability to benefit from the expanding knowledge of cellular phosphorylation is hampered.

Of the techniques that have been developed to interrogate kinase-phosphosite dependencies, most focus on identifying the targets of a kinase of interest; far fewer perform the similarly valuable task of identifying the kinase(s) that phosphorylate a particular residue within a substrate[9]. Indeed, we are not aware of an experimental method that has found widespread use that can selectively identify direct upstream kinases for any substrate. Broadly, existing approaches can be divided into three categories: (i) in silico predictions; (ii) screens in intact cells, and (iii) biochemical methods, often using cell extracts or recombinant kinases.

For a subset of kinases, consensus or optimal phosphorylation motifs have been determined[10–12]. A known phosphosite can be compared to these motifs to identify candidate kinases in silico[13]. However, such motifs have been produced for only about half of all human kinases and validated for far fewer. It is clear that false positives and negatives are frequent using prediction methods, even when combined with contextual information such as protein-protein interaction data, co-expression, or co-occurrence in literature abstracts[11,14–16].

The second category includes techniques such as kinome-wide RNAi, CRISPR/Cas9 or overexpression screens. Such screens are tremendously useful to biologists, but they often identify pathways or networks of kinases that are indirectly required for phosphorylation of a particular substrate rather than (or in addition to) the direct kinase[17–20]. Indeed, these indirect effects can make it difficult to identify the direct kinase for a particular phosphorylation site. For example, if RNAi for a kinase impedes cell cycle progression, it is difficult to determine whether its effect on a substrate is direct[21].

Finally, in the third category, a handful of approaches have been outlined. Arrays of recombinant kinases can be tested for their ability to phosphorylate substrates in vitro[22], but such libraries are incomplete, expensive, not widely available, and may not reflect the activity state of kinases in cells. Co-immunoprecipitation or other methods can be used to identify proteins that bind to substrates, but these techniques are not specific for kinases, nor for particular phosphosites, and kinase-substrate interactions may not be robust enough to be detected. Methods that make use of modified forms of substrates and/or ATP in an effort to crosslink substrates to their cognate kinases in cell extracts are promising. However, where tested, the specificity of kinase crosslinking has been low, these approaches require additional steps such as mass spectrometry to identify the crosslinked kinase[23–25], and they have not yet been used to identify unknown kinases. Tracking a specific kinase activity through biochemical enrichment steps can also be carried out, but this often requires large amounts of cell extract, multiple steps, and protein sequencing methods to identify candidate kinases[26–28].

Here, we develop a broadly applicable method to identify kinases for specific phosphorylation sites that overcomes many of these limitations. Our approach exploits the fact that, in recent years, large numbers of kinase inhibitors have been profiled for inhibitory activity on large panels of recombinant kinases[29–34]. Conventional wisdom suggests small molecule inhibitors are not appropriate tools for finding direct kinases because all such inhibitors inhibit more than one target, and signaling cascades in cells mean that indirect effects are common[29]. Our method, however, which we term kinase inhibitor profiling to identify kinases (KiPIK), circumvents these problems by treating the inhibition information for each recombinant kinase as a fingerprint for the identification of kinases acting on target phosphorylation sites in cell extracts. In this way, rather than ignoring or trying to minimize the influence of the off-target activity of inhibitors, the approach exploits this information. Here we describe the technique, validate it on diverse known kinase-phosphosite pairs, and use it to identify cellular kinases for as yet unassigned phosphosites on the Chromosomal Passenger Complex protein INCENP and the transcriptional activator BCL9L.

## Results

**Rationale for the KiPIK method.** KiPIK uses two key principles to allow identification of direct kinases for particular phosphorylation sites. First, it makes use of detailed selectivity profiles of small molecule kinase inhibitors obtained using large panels of recombinant kinases in vitro[31–35]. Together these provide unique inhibition patterns, or fingerprints, for ~80% of human protein kinases. Second, it utilizes the rapid action of kinase inhibitors in cell extracts to focus on defined biological states and to minimize indirect effects. The method relies on the finding that inhibition profiles of kinase inhibitors in cell extracts are generally similar to those established in vitro[36–38]. Like other approaches discussed earlier, it also rests on the assumption that kinase reactions in cell extracts preserve physiological kinase-phosphosite dependencies, a notion with strong anecdotal support, including the observation that extracts can carry out complex processes such as steps of the cell cycle[39–41].

In brief, KiPIK screening proceeds as follows (Fig. 1). Initially, cell conditions in which the phosphorylation of a protein residue of interest is robust are identified. For example, cells at a particular cell cycle stage can be enriched, or a signaling pathway leading to phosphorylation can be triggered. Whole-cell extracts

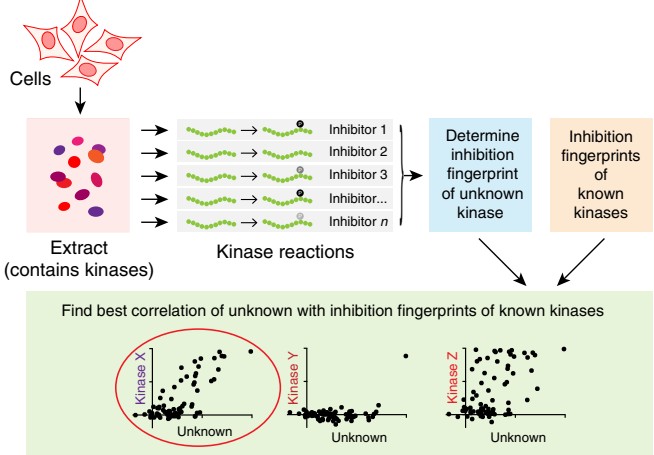

**Fig. 1 Outline of the KiPIK method.** In KiPIK, cell extracts (red box) are used as the source of all relevant kinases. Multiple kinase reactions are then carried out using an exogenous substrate of interest, each in the presence of single well-characterized kinase inhibitor (gray boxes) to produce an inhibition fingerprint for the unknown kinase (blue box). Comparison of this fingerprint with the known fingerprints of multiple kinases (orange box) using Pearson correlation coefficients (green box) allows candidate kinases to be ranked.

of these cells are prepared in the presence of phosphatase inhibitors to prevent the dephosphorylation of active kinases. These cell extracts are then used as the source of all potentially relevant kinases for in vitro kinase reactions using the substrate of interest. Multiple such in vitro kinase reactions are carried out in parallel, each in the presence of a member of a panel of well-characterized kinase inhibitors. This yields an inhibition fingerprint that characterizes the cellular kinase mainly responsible for the observed phosphorylation. Finally, to identify candidate kinases, this fingerprint is compared to the known inhibition patterns of all kinases in the available profiling datasets.

**The KiPIK method**. An informative KIPIK screen requires a library of kinase inhibitors for which in vitro inhibitory information data is available across a range of recombinant kinases. Suitable libraries include the protein kinase inhibitor sets (PKIS1 and PKIS2) which consist of published GlaxoSmithKline compounds selected by the Structural Genomics Consortium[34,35] and a library of inhibitors assembled by a team at EMD Millipore[32] and expanded at the Fox Chase Cancer Center[31]. The inhibitory properties of compounds in these libraries have been determined on large kinase panels in vitro using radiolabeled ATP incorporation assays[31,32], substrate electrophoretic mobility shift assays[34], or inhibitor binding assays[33–35]. Together, these profiles provide inhibition fingerprints for kinases encoded by 416 of the ~535 human protein kinase genes (Supplementary Data 1)[1,42].

A cell extract containing the phosphorylation activity being screened must also be prepared. In principle, any cell or tissue could be used as a source, but here we have focused on cultured cell lines. Prior to lysis, the cells can be treated to enhance the phosphorylation signal, or to probe phosphorylation dependencies in a particular context. For example, when investigating a mitotic phosphorylation, enriching the mitotic cell population by nocodazole treatment may be beneficial, while EGF stimulation can be used to explore phosphorylation downstream of EGFR. After lysis in the presence of phosphatase inhibitors, extracts are immediately flash frozen. A number of methods are available to assay substrate phosphorylation in cell extracts and a number of these are likely to be compatible with the KiPIK procedure. Here, we have used short biotinylated peptides encompassing phosphosites of interest as substrates, coupled with the use of phospho-specific antibodies to detect peptide phosphorylation in enzyme-linked immunosorbent assays (ELISA).

Using microplate robotics, many extract kinase reactions can be performed in parallel, each in the presence of a single kinase inhibitor from the characterized inhibitor library. Quantification of substrate phosphorylation in the presence of each inhibitor yields the required inhibition fingerprint that is characteristic of the predominant kinase acting on the substrate in the extract. This fingerprint is then compared with the known inhibition patterns previously determined for recombinant kinases challenged with the same set of kinase inhibitors. The kinase(s) with inhibition patterns showing the highest similarity to that of the unknown kinase activity (determined using Pearson's correlation coefficient, $\rho$) are the top hits in the screen.

**KiPIK for the histone H3S28 kinase identifies Aurora B**. To validate this methodology, we sought to identify the direct kinase responsible for histone 3 serine 28 (H3S28) phosphorylation in mitosis, previously reported to be Aurora B[43]. We made use of 312 kinase inhibitors from the PKIS1 panel (Supplementary Data 2) that have been profiled in vitro on two panels of recombinant kinases: a Nanosyn assay assessed inhibition of 203 kinases at inhibitor concentrations of 0.1 μM and 1 μM, whereas a DSF (differential scanning fluorimetry) assay assessed inhibitor

binding at 1 μM to 68 kinases by thermal denaturation[34]. Together, this provided coverage of 236 kinases (encoded by 234 unique genes). Extracts of HeLa cells that had been arrested in mitosis for 12 h in nocodazole were used as the source of relevant kinases. To create an inhibition fingerprint for the H3S28 kinase, the phosphorylation of an H3(21–44)-GK-biotin peptide was assayed using anti-H3S28ph antibodies in the presence of each inhibitor individually at 10 μM. For the H3S28 kinase fingerprint, the top three correlation scores were all from the Nanosyn 1 μM profiling dataset, and all were with Aurora family kinases (Fig. 2a; Supplementary Fig. 2). The highest overall correlation ($\rho = 0.74$) was with Aurora B. Notably, the top three hits within the Nanosyn 0.1 μM dataset were also all Aurora kinases. The DSF dataset did not contain the Aurora kinases, and no kinase in this dataset had a correlation score above 0.52. Therefore, the KiPIK method unambiguously identified Aurora family kinases as those most likely responsible for H3S28 phosphorylation in mitosis. Aurora B, the established kinase, was the top hit.

**KiPIK screens for tyrosine phosphorylation sites**. We wished to validate KiPIK screening for diverse kinases. We, therefore, conducted further screens to identify kinases for two tyrosine residues. The first was Y795 in the cytoplasmic tail of integrin β1A, thought to be phosphorylated by Src family kinases[44], and the second was the Y1016 autophosphorylation site in the carboxy-terminal tail of EGFR (Y992 in mature EGFR)[45,46]. In both cases, we used generic phospho-tyrosine antibodies to detect phosphorylation of biotinylated substrate peptides in extracts of A431 cells treated for 5 min with 50 ng/ml EGF, and the PKIS1 library of kinase inhibitors.

In the screen for kinases that phosphorylate integrin β1A Y795, the top 6 hits, and 10 of the top 11 overall, were members of the Src family kinase family (Fig. 2b, Supplementary Fig. 3). Therefore, this screen unambiguously identified members of the Src family as candidate kinases responsible for integrin β1 Y795 phosphorylation, strongly supporting previous suggestions. In the screen for kinases that phosphorylate residue Y1016 of EGFR, there were three clear candidates (Fig. 2c, Supplementary Fig. 4). The top hit was EGFR, and the two EGFR-related kinases ERRB2/HER2 and ERRB4/HER4 were second and third. Therefore, the KiPIK method successfully identified EGFR as the top candidate kinase for autophosphorylation of Y1016.

These results demonstrate that, for peptides containing a single tyrosine, generic phospho-specific antibodies can be used effectively for KiPIK screening, avoiding the need for phosphosite-specific antibodies. They also lend further support to the idea that the method can be used successfully to identity kinases for a variety of specific phosphorylation sites, including on receptor and non-receptor tyrosine kinases as well as on a mitotic serine/threonine kinase.

**KiPIK screening preferentially identifies direct kinases**. To substantiate the idea that KiPIK screens are less susceptible to indirect upstream kinase activity than genetic screens, we turned to a kinase that we know is under the control of an upstream kinase (Aurora B) that is active in KiPIK conditions. Haspin is the direct kinase that phosphorylates Histone H3T3 during mitosis[47]. However, in cells, Aurora B inhibition also reduces H3T3ph because Aurora B activates Haspin and inhibits the RepoMan-PP1 phosphatase that dephosphorylates H3T3ph[41,48]. In contrast, in cell extracts containing phosphatase inhibitors, Aurora B inhibitors do not compromise generation of H3T3ph[41], presumably because Haspin is already phosphorylated and activated, and phosphatases are inactive. Consistent with this, when we carried out a kinome-wide RNAi screen to identify kinases

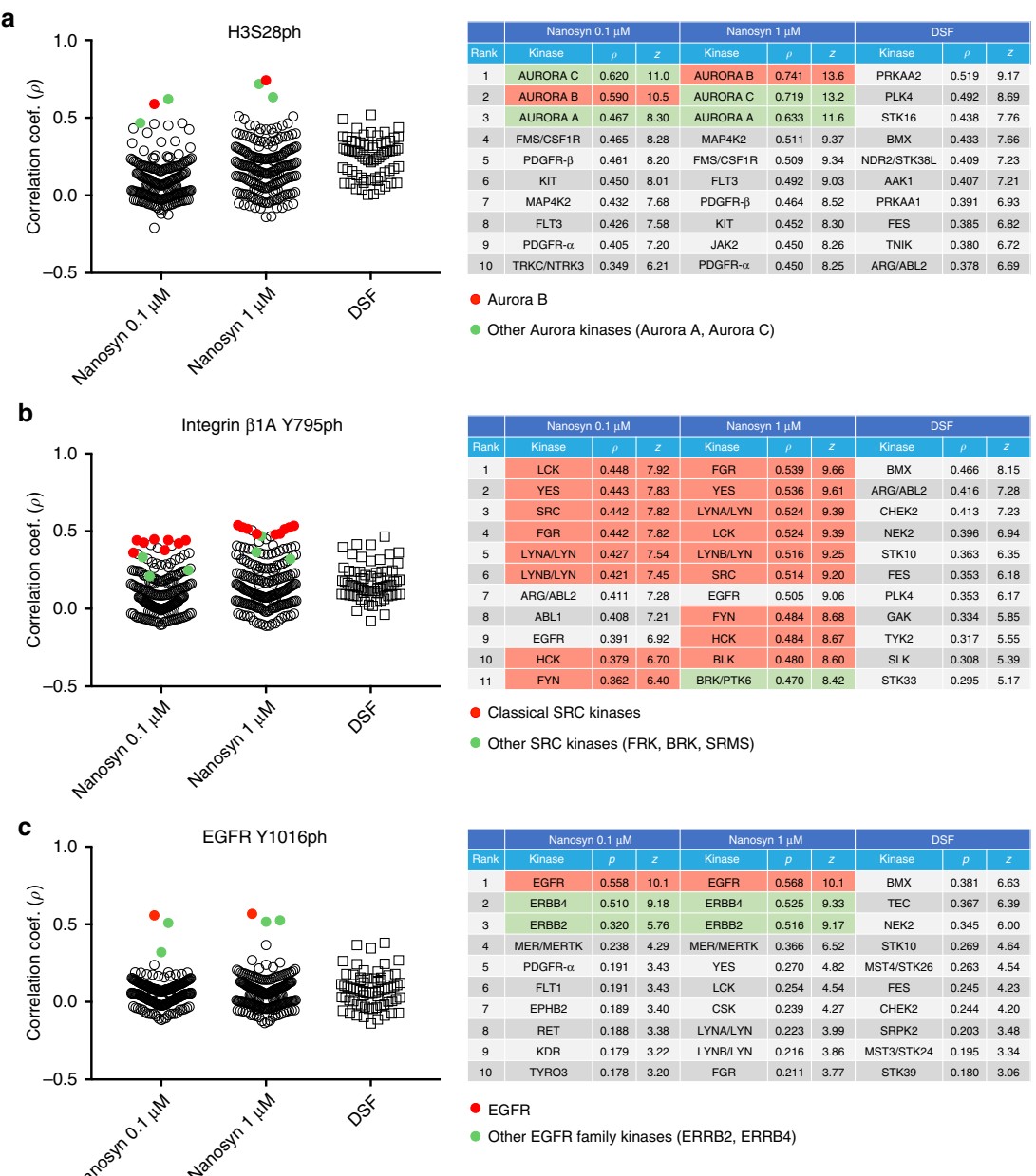

**Fig. 2 Results of KiPIK screens using the PKIS1 inhibitor library. a** H3S28ph, **b** Integrin β1A Y795ph, and **c** EGFR Y1016ph. In each case, stripcharts on the left show the Pearson correlation coefficients between the %inhibition results from the KiPIK screen and the % inhibition results for each of the kinases in the Nanosyn profiling datasets at 0.1 and 1 μM, and the DSF profiling dataset. The tables on the right show the top 10 (or 11) hits from each profiling dataset with corresponding correlation coefficients ($\rho$) and z-scores. Expected kinase(s) are shown in red, and closely related kinases in green. Note that Aurora kinases are not present in the DSF dataset, and Src and EGFR-related kinases are not present in the Nanosyn datasets. Source data are provided as a Source Data file.

required for H3T3ph in mitotic HeLa cells, Haspin was the top hit, but Aurora B kinase was also identified in the top 3 (Fig. 3a). However, when we conducted a KiPIK screen using a H3(1–21) peptide to identify the kinase responsible for phosphorylating H3T3 in mitotic HeLa extracts (Fig. 3b, Supplementary Fig. 5), none of the Aurora family kinases was within the top 40 (highest $\rho = 0.32$), while Haspin was the top hit ($\rho = 0.66$). In addition, the kinase Plk1, which also has a role in activating Haspin in cells[49,50], was in the top 5 protein kinase hits in the RNAi screen but was in the bottom 20% of the KiPIK screen. Therefore, when compared to an RNAi screen, a KiPIK screen for H3T3ph was relatively insensitive to the kinases upstream of the direct kinase.

EGFR and Src family kinases participate in intimate crosstalk in A431 cells. Src family kinases are activated in response to

EGFR signaling in these cells, and EGFR inhibitors prevent Src activation[51–53]. In addition, active Src family kinases can feed back to activate EGFR[54–56]. When we conducted KiPIK screens in EGF-stimulated A431 extracts for the Src-family substrate integrin β1A Y795, EGFR was present among the top ten kinases (Fig. 2b). Similarly, several Src family kinases were found in the top twenty hits when we conducted a KiPIK screen for the EGFR autophosphorylation site on Y1016 (Fig. 2c). Nevertheless, as described earlier, KiPIK unambiguously identified Src family kinases as the top candidates for integrin β1A Y795 phosphorylation, and EGFR as the top candidate for autophosphorylation at Y1016.

Together, these results demonstrate that KiPIK screening identifies the direct kinase responsible for a particular

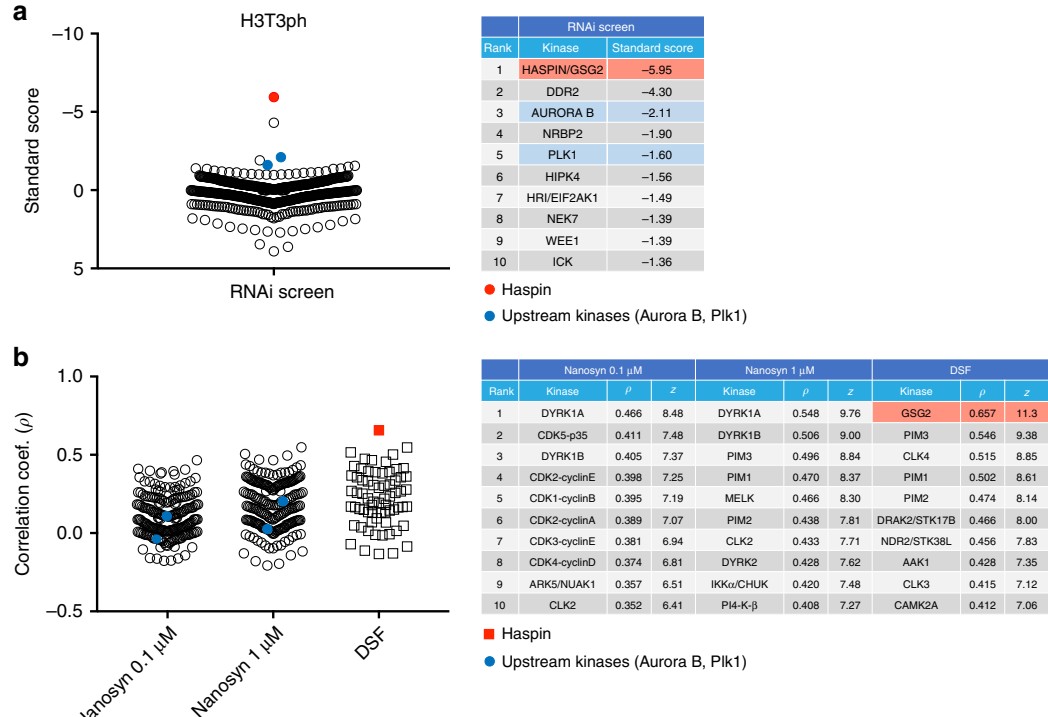

**Fig. 3 Results of RNAi and KiPIK screens for H3T3ph. a** A kinome-wide RNAi screen for H3T3ph in mitosis was carried out in HeLa cells using immunofluorescence and High Content Analysis. The stripchart on the left shows the standard score (proportional to staining intensity) for each protein kinase in the library. The table on the right shows the top 10 protein kinase hits from the screen with corresponding standard scores. **b** Results of a KiPIK screen using the PKIS1 inhibitor library for H3T3ph. The stripcharts on the left show the Pearson correlation coefficients with each kinase in the Nanosyn profiling datasets at 0.1 and 1 μM, and the DSF profiling dataset. The table on the right shows the top 10 hits from each profiling dataset with corresponding correlation coefficients and z-scores. The expected kinase (Haspin/GSG2) is shown in red, and kinases that are known to indirectly modulate H3T3ph are shown in blue. Note that Haspin/GSG2 is not present in the Nanosyn datasets. Source data are provided as a Source Data file.

phosphorylation event and is relatively insensitive to the indirect effect of kinases operating in signaling networks.

**KiPIK identification of a kinase for INCENP S446**. We next sought to use KiPIK identify the kinase for an unassigned phosphosite. We were intrigued by consensus motifs, identified using mass spectrometry by Dephoure et al.[4], which are commonly phosphorylated in mitosis but for which the responsible kinase(s) were not identified. For one of these motifs, P-X-pS-X-X-[K/R], we identified a site in the Inner Centromere Protein (INCENP) for which a phosphospecific antibody had been raised: INCENP S446ph[57]. Phosphorylation of this site was previously shown to be sensitive to the Aurora B inhibitor Hesperadin[57], but this inhibitor is relatively non-selective[58] and the site does not match the classical Aurora kinase consensus. To identify the kinase responsible for phosphorylation of this site, we conducted a KiPIK screen using a biotin-INCENP(439–453) peptide (GPREPPQ**S**ARRKRSY), and extracts of mitotic HeLa cells as a source of kinases. This screen unambiguously identified Cyclin-dependent kinases as the best hits in all three profiling datasets. The master regulator of mitosis, Cyclin B1-Cdk1, was the top hit ($\rho = 0.81$; Fig. 4a, Supplementary Fig. 6).

To confirm that Cyclin B1-Cdk1 was a bona fide kinase for INCENP S446, we first conducted in vitro kinase assays. Indeed, Cyclin B1-Cdk1 efficiently phosphorylated S446 on a recombinant INCENP fragment containing residues 369–583, but not the TSS motif (S893/S894) in recombinant INCENP residues 825–918. In contrast, Aurora B was able to phosphorylate its known target, the TSS motif of INCENP, but not S446 (Fig. 4b).

Full length recombinant INCENP was also phosphorylated by Cyclin B1-Cdk1 in vitro (Supplementary Fig. 7A). Furthermore, INCENP immunoprecipitated from nocodazole-arrested mitotic cells was recognized by the INCENP S446ph antibody, and this phosphorylation was sensitive to the Cdk inhibitor Roscovitine (Supplementary Fig. 7B). However, determining the direct substrates of Cyclin B1-Cdk1 in cells is complicated because inhibition of Cdk1 prevents mitotic entry and/or forces mitotic exit, leading to indirect effects on multiple kinase substrates. To address this, we used immunofluorescence microscopy and high content imaging to analyze the kinetics with which INCENP S446ph was lost following acute Cdk inhibition in nocodazole-arrested mitotic cells. The proteasome inhibitor MG132 was also included because it prevents the degradation of key mitotic regulators such as Cyclin B1 and, in some instances, is reported to delay mitotic exit even in the presence of Cdk1 inhibitors[59]. Aurora B relocates to the spindle midzone at the end of mitosis so, as an additional way to focus on cells still in mitosis, we analyzed only cells in which Aurora B remained on chromosomes. We found that Cdk1 inhibition using 10 μM RO-3306 led to a 50% reduction in intensity of INCENP S446ph over 9 min, while the Aurora B phosphorylation site INCENP TSS was unaffected (Fig. 4c). In contrast, inhibition of Aurora B with 10 μM ZM447439 led to a rapid decline of INCENP TSSph but did not influence INCENP S446ph over the same time period (Fig. 4c). Also consistent with Cyclin B1-Cdk1 being the kinase responsible, INCENP S446ph was detected on chromosomes in mitotic cells from early mitosis until metaphase when Cyclin B1-Cdk is active, but was lost at anaphase when Cyclin B is degraded (Fig. 4d). This pattern mirrored that of previously reported Cyclin

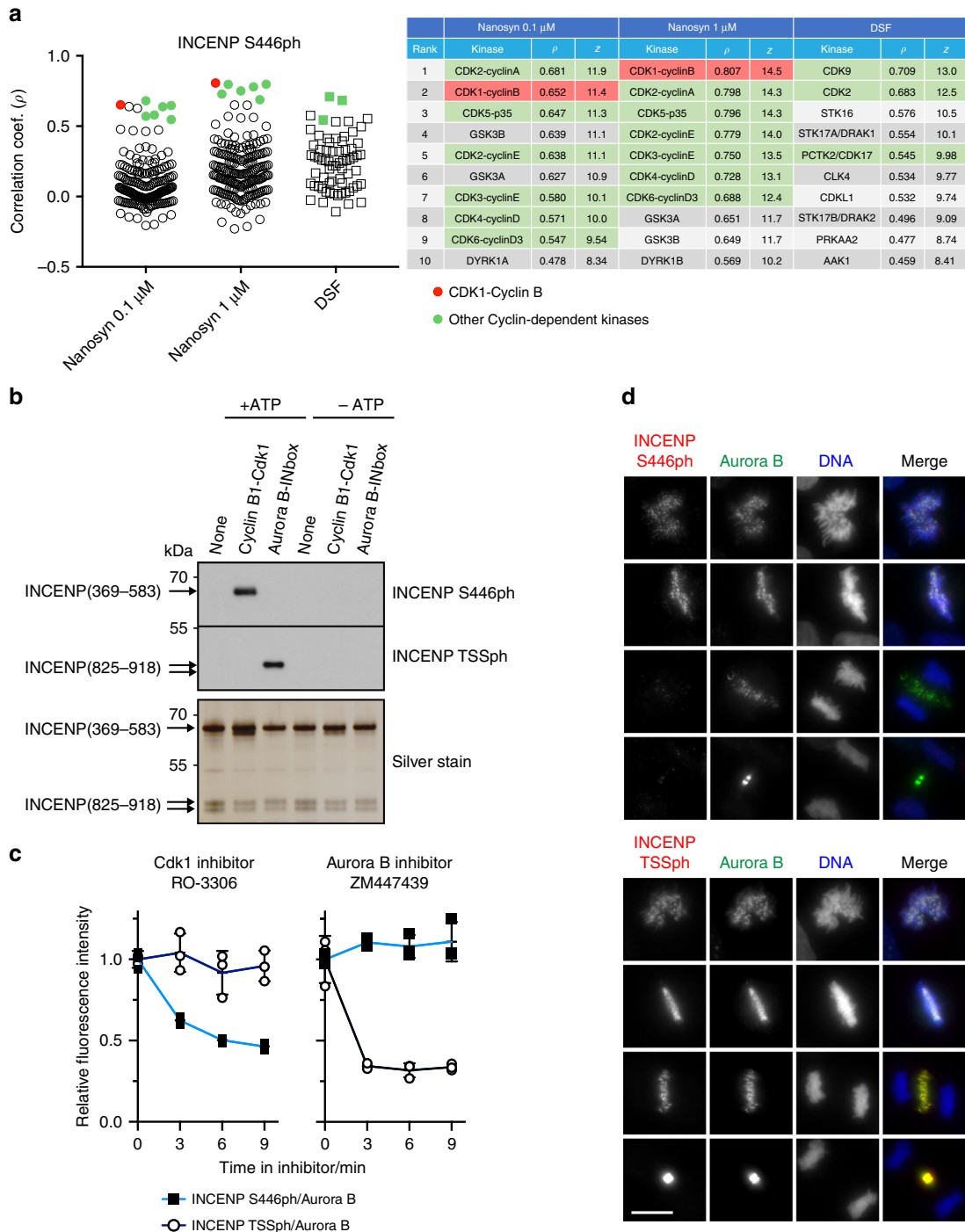

**Fig. 4 Results of a KiPIK screen for INCENP S446ph, and confirmatory experiments. a** The stripcharts on the left show the Pearson correlation coefficients with each kinase in the Nanosyn profiling datasets at 0.1 and 1 μM, and the DSF profiling dataset. The table on the right shows the top 10 hits from each profiling dataset with corresponding correlation coefficients and z-scores. The overall top hit kinase is shown in red, and closely related kinases in green. Note that Cdk1 is not present in the DSF dataset. **b** Recombinant Cyclin B-Cdk1 efficiently phosphorylates INCENP S446, but not INCENP TSS, in vitro. Recombinant INCENP(369–583) and INCENP(825–918) were provided as potential substrates for GST-CDK1/GST-CycB or Aurora B/INbox and phosphorylation was detected using the corresponding phosphospecific antibodies. The experiment was done twice with similar results. **c** Acute inhibition of Cdk1, but not Aurora B, lowers INCENP S446ph (black squares) but not INCENP TSSph (open circles) in mitotic HeLa cells. In contrast, acute inhibition of Aurora B lowers INCENP TSSph, but not S446ph. The y-axis shows the ratio of INCENP phosphorylation/Aurora B intensity on chromatin. Only cells in which Aurora B was on chromatin were analyzed. Error bars show means ± SD (n = 3 separate cell cultures treated in parallel). **d** Immunofluorescence microscopy of INCENP S446ph and INCENP TSSph in mitotic HeLa cells. Both INCENP S446ph and TSSph are phosphorylated in early mitosis but, after INCENP and Aurora B move to the spindle midzone in anaphase cells, INCENP is phosphorylated at the TSSph Aurora B target residues but not at the Cdk1 target S446ph. Scale bar is 10 μm. The experiment was done twice with similar results. Source data are provided as a Source Data file.

B1-Cdk1 target sites on INCENP, but was distinct from that of INCENP TSSph (Fig. 4d) which is generated by Aurora B activity throughout mitosis, including on the central spindle in anaphase[60]. Together, these results show that Cyclin B-Cdk1 is the primary direct kinase for INCENP S446 in cells, and that KiPIK can identify the kinases required for the generation of orphan phosphorylation sites.

**Inhibitor library requirements for KiPIK screening**. We have shown that a single inhibitor panel, PKIS1, is suitable for use in KiPIK screening. We wished to determine if additional inhibitor panels and profiling datasets could be utilized. Where alternative inhibitor panels have been profiled on additional kinases, this allows the number of kinases covered by KiPIK screening to be increased. Furthermore, not all kinase inhibition profiles are necessarily concordant between different profiling efforts[61], and the extent to which different compound sets give similar results provides an indication of the robustness of the method.

To explore this, we assembled a custom library of 128 inhibitors that have been used in other kinase profiling studies (Supplementary Data 3). This included all 72 inhibitors profiled on 388 different kinases by Davis et al. ("Davis") using the DiscoverX competition binding assay[33], 76 of the 178 inhibitors profiled at 0.5 μM on 300 kinases by Anastassiadis et al. ("Anastassiadis") using conventional in vitro kinase assays at a fixed concentration of 10 μM ATP[31], and 55 of the 158 inhibitors profiled at 1 μM or 10 μM on 234 kinases by Gao et al. ("Gao 1 μM" and "Gao 10 μM"), also using in vitro kinase assays, but at near-$K_m$ concentrations of ATP[32]. Together, this encompasses 431 kinases (405 unique human kinase genes; Supplementary Data 1).

Using this library, we repeated KiPIK screens for mitotic H3T3 and H3S28 phosphorylation. For H3T3ph, the highest overall correlation coefficient ($\rho = 0.69$) was with Haspin in the Gao 1 μM dataset, and Haspin was also the top scoring kinase ($\rho = 0.66$) in the Anastassiadis dataset. Haspin was not present in the Davis dataset, and no kinase in this panel had a correlation coefficient exceeding 0.52 (Fig. 5a). As before, the upstream regulatory kinases for Haspin, Plk1 and Aurora B, did not score highly in the screen (Fig. 5a). For H3S28ph, the highest overall correlation coefficient ($\rho = 0.78$) was with Aurora B in the Davis dataset, and Aurora B was also the top scoring kinase ($\rho = 0.66$) in the Anastassiadis dataset (Fig. 5b). Therefore, KiPIK screening with alternative inhibitor and kinase panels was able to accurately identify the appropriate kinase for both sites.

In the H3S28ph screen, the three Aurora kinases were all in the top 25 kinases in the Gao 1 μM dataset, but they were not ranked as highly in the Gao 10 μM dataset. In the H3T3ph screen, Haspin was the third highest-scoring kinase in the Gao 10 μM dataset. One likely reason for poorer performance in these three cases was apparent when the KiPIK correlation plots for Haspin and Aurora B in the Gao datasets were examined (Supplementary Fig. 8): it was clear that more of the Haspin and Aurora B inhibitors in the Gao datasets caused near 100% kinase inhibition than in our KiPIK screening data. This decreased the linearity of the correlation plots, and so reduced the correlation coefficients. This highlights the importance of using suitable inhibitor concentrations in both KiPIK and profiling studies.

A second likely reason for the varying utility of profiling datasets was the different number of inhibitors we tested from each (Fig. 5; 310 for Nanosyn, 311 for DSF, 76 for Anastassiadis, 72 for Davis, 55 for Gao). To more formally test the impact of inhibitor panel size, for those KiPIK screens where a clear single kinase was identified as the expected top hit (For H3T3ph: DSF, Anastassiadis, Gao 1 μM; for H3S28ph: Nanosyn 1 μM, Anastassiadis, Davis; and for EGFR Y869ph: Nanosyn 0.1 μM and 1 μM),

we downsampled the inhibitor panel size by computationally removing random but fixed numbers of inhibitors. This analysis revealed that, when using large inhibitor panels, the identification of the correct kinase was generally robust until the inhibitor library size decreased below about 100 inhibitors (Supplementary Figs. 9, 10). Consistent with this, for the smaller panels of inhibitors, identification of the correct kinase fell off quickly as the numbers of inhibitors used decreased. It is therefore unsurprising that the smallest inhibitor panel used (for the Gao dataset) is the least robust. Nevertheless, the results suggest that the use of more inhibitors is likely to increase the utility of these additional datasets.

Given these considerations, it would be useful to have a simple metric to allow the robustness of different screens to be compared, taking into account the number of inhibitors used and the strength of observed correlation coefficients. We estimated the null distribution of correlation coefficients for each screen by random permutation of % inhibition results (Supplementary Fig. 11A), and then calculated z-scores for each experimentally-determined correlation coefficient, reflecting how many standard deviations each is from the mean of the null distribution (Supplementary Fig. 11B, C; z-scores are also provided in Figs. 2–6). While the z-score does not correct for all the factors contributing to inter-assay variation, it appears that screens in which the top z-scores are below 6 should be treated with caution, while z-scores of 10 or above appear robust.

**An alternative inhibitor library identifies a BCL9L kinase**. While this work was in progress, a new set of 485 inhibitors, PKIS2, profiled at 1 μM using the DiscoverX assay became available[35]. The dataset covers 403 human kinases encoded by 389 unique genes. We used this new library to identify the kinase for an unassigned phosphorylation site. Specifically, we examined phosphorylation of S915 in the transcriptional activator BCL9L (B-cell lymphoma 9-like protein), a phosphosite identified by mass spectrometry among proteins enriched using a phospho-AMPK substrate motif (L-X-R-X-X-[pS/pT]) antibody[62]. BCL9L S915ph is upregulated by phorbol ester treatment (see Cell Signaling Technology; https://www.cellsignal.co.uk/products/primary-antibodies/phospho-bcl9l-ser915-antibody/13325), suggesting that it lies within a PKC-triggered signaling pathway. To identify the relevant direct kinase, we conducted a KiPIK screen using 399 PKIS2 inhibitors, a biotin-BCL9L(908–922) peptide, extracts of TPA-treated HeLa cells as a source of kinases, and an anti-BCL9L S915ph antibody. This screen clearly identified the AGC kinase family as the source of the best hits (the top 20 are all AGC kinases). The top hit was the cAMP-dependent kinase PKA-β ($\rho = 0.87$) and PKA-α was also within the top ten, as were PKC-δ, -η and -ε. To test the robustness of the screen, we repeated it using all 485 available PKIS2 inhibitors and PKA-β was again the top hit ($\rho = 0.77$; Fig. 6a, Supplementary Fig. 12).

Further experiments confirmed that: (i) RNAi of PKA-α/β diminished BCL9L S915 phosphorylation in TPA-stimulated HeLa cells (as did RNAi of phorbol-ester-sensitive PKC isoforms, as expected; Fig. 6b); (ii) Forskolin, which activates PKA by stimulating cAMP production[63], increased BCL9L S915ph in cells (Fig. 6c), (iii) a small molecule that inhibits PKA, but not PKC (H-89)[29,31,32], decreased the phosphorylation of BCL9L S915 in cells and, as expected, a PKC inhibitor (Gö 6983)[31,32] also decreased BCL9L S915ph, while inhibitors with activity against AKT and LATS kinases (MK-2206 and SU-14813, respectively)[33,64] did not (Fig. 6d); (iv) BCL9L S915 peptides were more strongly phosphorylated by recombinant PKA-α than by PKC-δ in vitro (Fig. 6e); and (v) PKA-α but not PKC-δ phosphorylated full-length immunoprecipitated BCL9L S915 (Fig. 6f). Together, these results

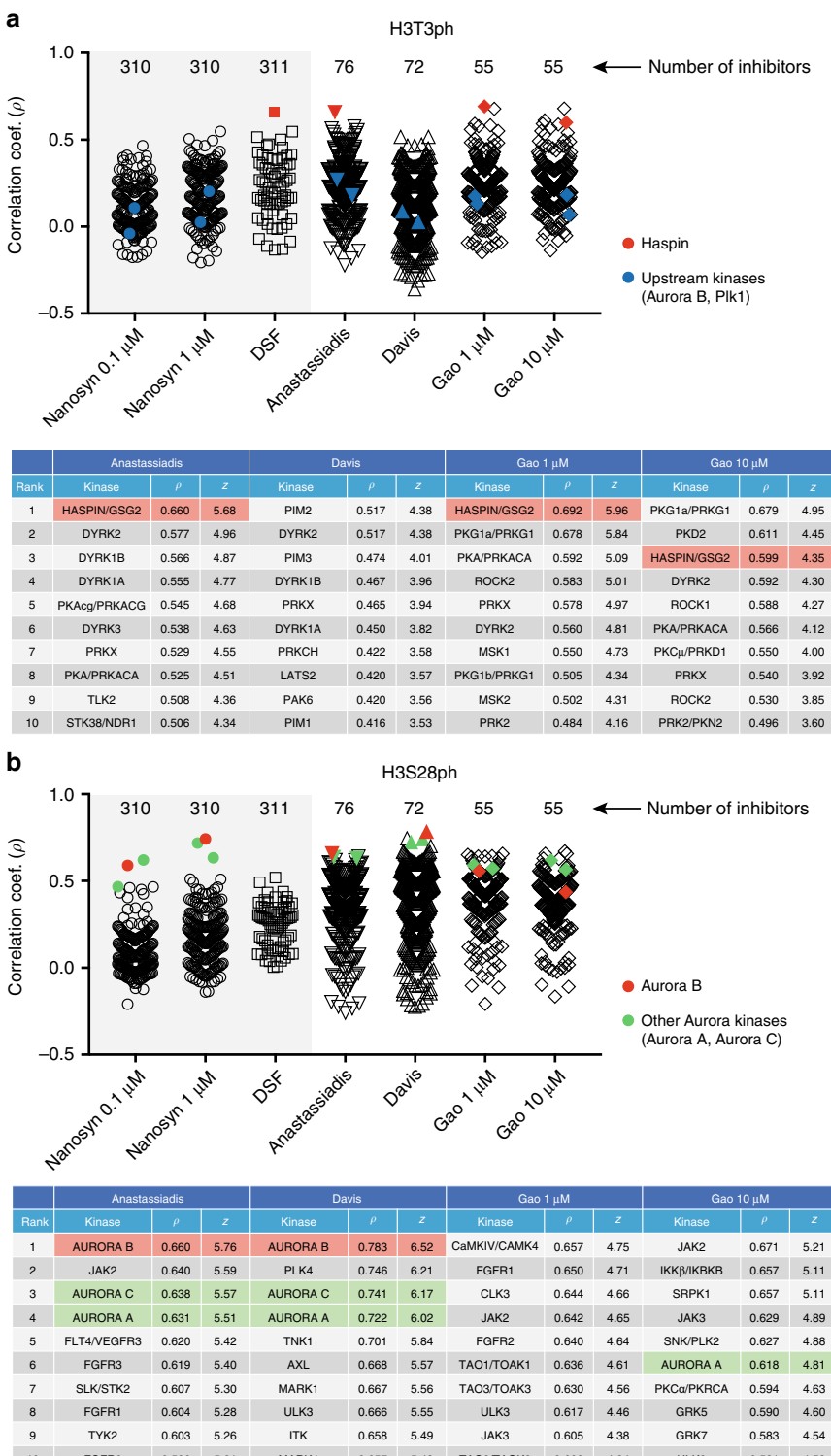

**Fig. 5 Results of KiPIK screens using the custom inhibitor library. a** KiPIK screen for H3T3ph. Stripcharts show the Pearson correlation coefficients of the %inhibition results from the KiPIK screen against the %inhibition results for each of the kinases in the Anastassiadis, Davis and Gao (1 and 10 μM) profiling datasets. Results from the PKIS1 screen using Nanosyn and DSF datasets are shown for comparison (shaded gray). The number of inhibitors included in each dataset is indicated above the plot. The table below shows the top 10 hits from each profiling dataset with corresponding correlation coefficients and z-scores. The expected kinase (Haspin/GSG2) is shown in red, and kinases that are known to indirectly modulate H3T3ph are shown in blue. **b** KiPIK screen for H3S28ph. Scatterplots and the table below are as indicated in (**a**). Expected kinase(s) are shown in red, and closely related kinases in green. Source data are provided as a Source Data file.

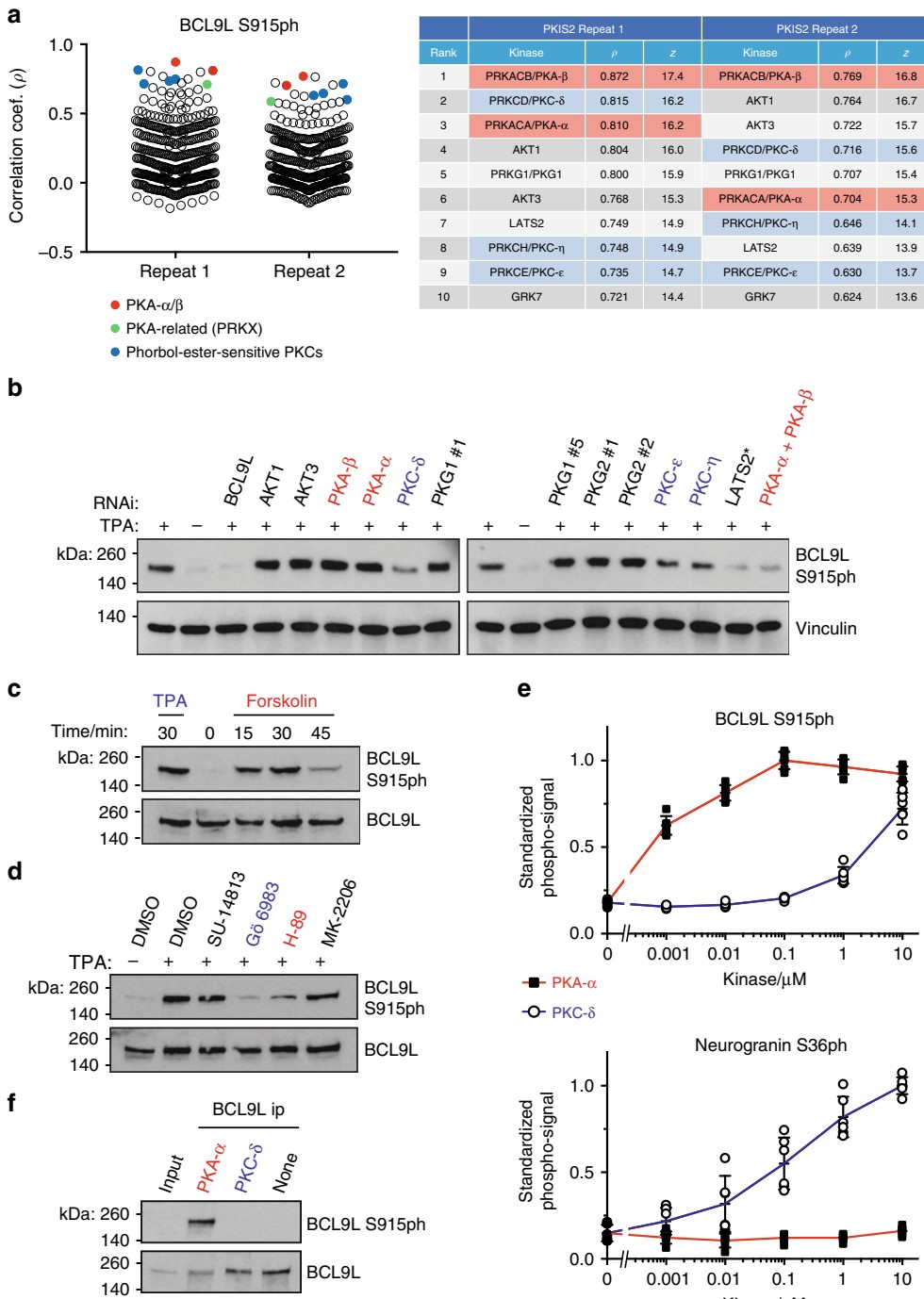

**Fig. 6 Results of KiPIK screens for BCL9L S915ph and confirmatory experiments. a** The stripcharts on the left show the Pearson correlation coefficients with each kinase in the PKIS2 DiscoverX profiling dataset, for repeats 1 and 2. The table on the right shows the top 10 hits with corresponding correlation coefficients and z-scores. The overall top hit kinase (PKA) is shown in red, and the closely related kinase PRKX in green. Phorbol-ester-responsive PKC kinases are shown in blue. **b** RNAi of top hit kinases shows that co-depletion of PKA-α and -β diminishes TPA-induced BCL9L S915ph in HeLa cells, as does depletion of PKC isoforms. Loss of BCL9L S915ph was seen following PKA and PKC depletion in three separate experiments, but loss of BCL9L S915ph following LATS2 RNAi (*) was not a reproducible finding. **c** Forskolin increased BCL9L S915ph in HeLa cells (one of two separate experiments shown). **d** Inhibitors of PKA and PKC, but not of LATS and AKT kinases, decrease BCL9L S915ph in HeLa cells. Similar results were obtained in a second experiment. **e** Recombinant PKA-α (red line, black squares), but not PKC-δ (blue line, open circles), efficiently phosphorylates BCL9L(908–922) peptides in vitro. In contrast, PKC-δ (but not PKA-α) phosphorylates its known substrate Neurogranin(28–43). Phosphorylation was detected using antibodies to BCL9L S915ph and Neurogranin S36ph. Error bars show means ± SD (n = 6 from two separate experiments). **f** BCL9L immunoprecipitated from HeLa cells is phosphorylated in vitro by recombinant PKA-α, but not PKC-δ. The experiment was done twice with similar results. Source data are provided as a Source Data file.

indicate that KiPIK using the PKIS2 library correctly identified PKA as the direct BCL9L S915 kinase in HeLa cells, even where the apparent upstream kinase PKC was strongly activated by phorbol ester treatment.

**Alternative approaches to analyse correlation data.** In cases where the kinase of interest is present in the profiling dataset, we have shown that KiPIK is able to identify the correct kinase or kinase sub-family in the six cases we have tested. However, because only 80% of known protein kinases are covered by the profiling data it is possible, for some screens, that the unknown kinase is not present in the profiling data. Nevertheless, the top hits are still likely to share properties with the unknown kinase, and therefore can help prioritize candidate kinases for further study. As we have shown above, a simple ranking of correlation coefficients readily identifies relevant candidates, but we explored two additional approaches to enhance this analysis.

First, the well-known sequence-based kinome dendrogram of Manning et al.[42] can be annotated with correlation coefficients derived from KiPIK screening (Figs. 7 and 8, and Supplementary Figs. 13–20). This allows kinases that are closely related in sequence to the top hits in the screen to be easily located, and sub-families of kinases for further testing can then be identified. For example, the DSF profiling set does not contain the kinase Cdk1, but strong hits on the cyclin-dependent kinase (CDK) sub-branch of the CMGC kinase family clearly highlight the CDK sub-family as a source of excellent candidates for the INCENP S446ph kinase (Fig. 8a, Supplementary Fig. 19).

Second, hierarchical clustering of all kinases based on the similarity of their inhibition fingerprints can be carried out (using Pearson correlation coefficients). This generates kinome trees in which kinases are clustered more closely if they are inhibited by similar sets of inhibitors (Supplementary Figs. 21–28). As previously described, such inhibition-based trees have general similarities to sequence-based trees, but also notable differences, such as separating the p38α and p38β kinases from the closely-related p38δ and p38γ kinases[58,65]. A caveat of this approach is that, although bootstrapping gives an indication of the reliability of the branching pattern (Supplementary Figs. 29–32), a kinase can only be in one place in a tree, so alternative possible kinase clusters are obscured. Nevertheless, if the inhibition fingerprint of the unknown kinase is included in the clustering process, then kinases that have similar inhibition profiles, but may be distant in sequence relatedness, can be identified. For example, Haspin is not present in the Davis profiling set, but the inhibition fingerprint of H3T3ph kinase activity clusters in a group that contains DYRK2 (Supplementary Fig. 22A). This kinase is a common off-target of Haspin inhibitors[66–68] and it clusters close to Haspin in profiling sets which do contain both Haspin and DYRK2 (ie Anastassiadis and Gao 1 μM; Supplementary Fig. 22B). In this way, information derived from the properties of kinases and inhibitors that were not included in the experimental screening process can be exploited to identify kinases for further testing.

## Discussion

Here we describe a method for identifying the kinases for specific protein phosphorylation sites that outperforms in silico and genetic screening approaches in key respects. KiPIK identified the previously determined kinase for all examples we tested, including serine/threonine kinases that act on histones in mitosis, and both receptor and non-receptor tyrosine kinases. We also tested the method on two orphan phosphorylation sites. For a residue in the Chromosomal Passenger Complex protein INCENP (S446) that sits within a consensus motif commonly phosphorylated in

mitosis by unknown kinase(s)[4,57], KiPIK unambiguously identified Cyclin-dependent kinases as candidates, and Cyclin B1-Cdk1 was the top hit. Follow up studies supported the view that this kinase is responsible for INCENP S446ph in cells. Together, these results suggest that the P-X-S-X-X-[K/R] motif is a common non-canonical target for Cdk1 action in mitosis, an idea supported by findings that such motifs are favorable in vitro targets for Cyclin B-Cdk1[69]. S446 lies within a region of INCENP (PRD1) that has recently been shown to be heavily phosphorylated by Cyclin B1-Cdk1 on other residues, and therefore may be an additional site that modulates the association of INCENP with microtubules[70]. We also found that, after phorbol ester treatment of cells, PKA can phosphorylate S915 of BCL9L. This is consistent with findings that phorbol esters can trigger PKA activity, perhaps via the phosphorylation of adenylate cyclase by PKC[63]. BCL9L functions in the Wnt signaling pathway by interacting with β-catenin to promote TCF-mediated transcription[71,72]. Our findings suggest that it may be fruitful to investigate if phosphorylation of BCL9L by PKA, as well as the reported phosphorylation of β-catenin by PKA[73,74], underlies crosstalk between PKA and Wnt signaling[75].

KiPIK has a number of advantages over current methods for identifying kinases for phosphorylation sites. For example, biochemical approaches that exploit substrate-kinase interactions or attempt kinase purification by tracing enzyme activity require large samples and customized methods that are not easily transferable to additional substrates. In contrast, KiPIK is a relatively standardized approach that can be applied to a variety of substrates. Methods such as RNAi and CRISPR/Cas9 (or overexpression) can be used, but they require long timeframes, causing indirect effects and limiting analysis at different stages of a process such as the cell cycle. These methods also require efficient transfection, and genetic manipulation prohibits simple analysis of the latter stages of a cellular or developmental process if there is a defect in an earlier step. Small molecule inhibitors, in contrast, act in minutes and can be used to determine the effect of kinase inhibition on a particular biological state without affecting a preceding step. A recently published approach using inhibitor panels in intact cells appears useful, but it identifies networks of kinases upstream of substrate phosphorylation events[76]. Our method differs substantially from this because it utilizes extracts of cells harvested in a state where signaling pathways are already active and the presence of phosphatase inhibitors minimizes the requirement for the ongoing activity of upstream kinases, biasing KiPIK towards identification of the direct kinases for particular substrate residues. This allows, for example, the successful identification of Cyclin B1-Cdk1 as a direct kinase for INCENP S446 kinase, even though loss of Cdk1 prevents cells from entering mitosis[21], and allows the direct and indirect contributions of Haspin and Aurora B to H3T3ph to be uncoupled. Inhibitors also allow analysis of a broad range of cells and tissues which may be difficult to transfect. In addition, numerous kinases have kinase-activity-independent functions, and inhibitors provide a powerful way to focus on the function of kinase activity. In practice, however, many kinase inhibitors are poorly selective, and this significantly hampers their use in conventional screening approaches. In KiPIK screening, however, we harness knowledge of this incomplete selectivity to enhance the information content of inhibitor screens to identify candidate kinases for particular phosphorylation sites.

KiPIK has a number of theoretical advantages over in silico approaches. In particular, it utilizes cell extracts and so provides biological context that is absent from most prediction methods. For KiPIK, cells can be treated prior to extract preparation to ensure that the relevant kinase and its regulatory subunit(s) are present and active, and over 400 human kinases are currently covered by the method. Kinases that are not expressed in the

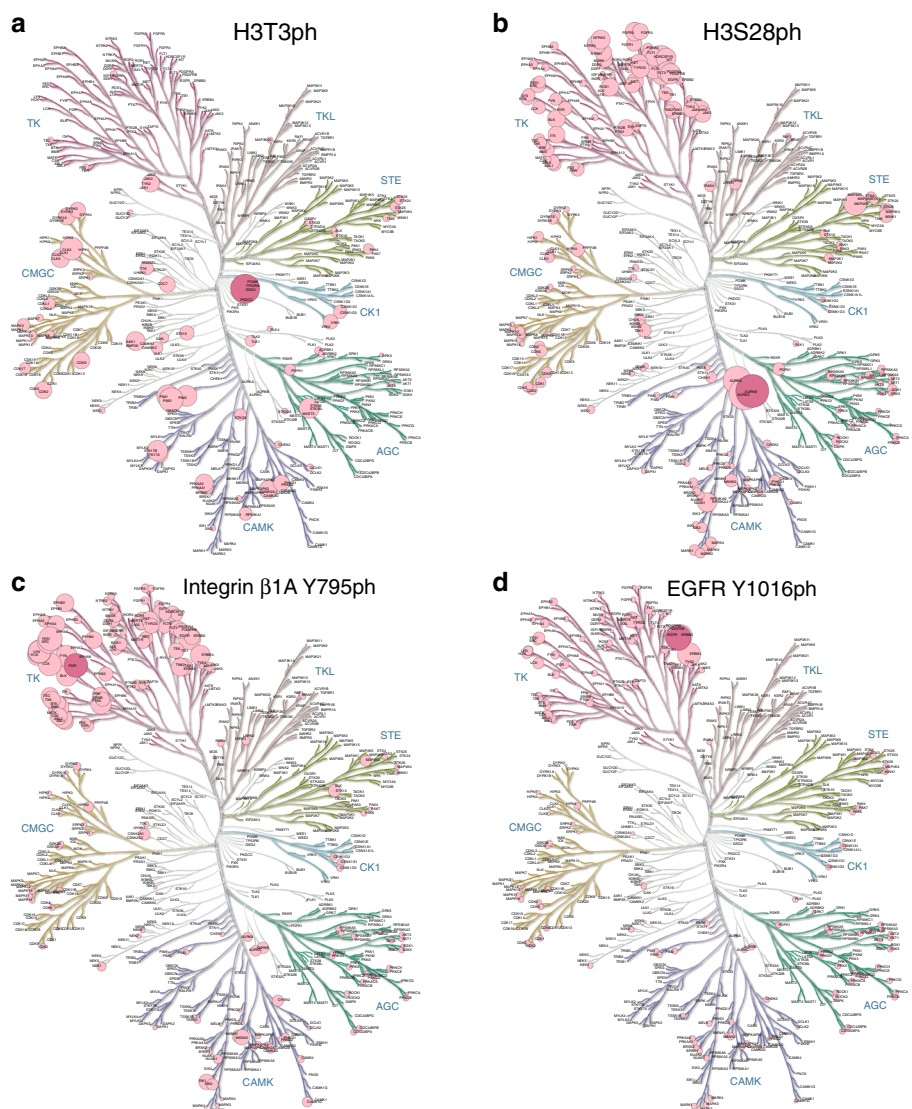

**Fig. 7 KiPIK correlation coefficients for known kinases displayed on human kinome dendrograms. a** H3T3ph results from the PKIS1/DSF dataset. **b** H3S28ph results from the PKIS1/Nanosyn 1 μM dataset. **c** Integrin β1 A Y795ph results from the PKIS1/Nanosyn 1 μM dataset. **d** EGFR Y1016ph results from the PKIS1/Nanosyn 1 μM dataset. Circle sizes indicate correlation coefficients (from no circle at $\rho = 0$ to the largest circle at $\rho = 1$). The expected kinase, and overall top hit in each screen, is shown in deep violet. Trees for all results can be found in Supplementary Figs. 13–18. Kinome trees were produced using KinMap[92], and the underlying tree illustration is reproduced courtesy of Cell Signaling Technology, Inc. (www.cellsignal.com). Source data are provided as a Source Data file.

relevant cell are excluded, and only kinases that efficiently phosphorylate the substrate in competition with other kinases and substrates in the extract will be identified. In contrast, the majority of computational methods are limited in the number of kinases for which they can make predictions, and they do not account for factors such as the activation state or expression level of the kinase[14]. Indeed, compared with a spectrum of in silico prediction methods, including those that attempt to integrate information on biological context, KiPIK was the most reliable approach for identifying the kinases for the six substrates examined here (see Supplementary Fig. 33).

As with all screening approaches, there are limitations of the KiPIK method. Although the inhibitor libraries we have used contain inhibitors for the great majority of kinases[31,32,34], we cannot be sure that all kinases are equally discoverable. For an individual screen, KiPIK provides an unambiguous ranking of candidate kinases that can be followed up with additional

experiments. In addition, z-scores provide an indication of the robustness of the results. Screens in which all z-scores are below 6 may be questionable, but future utilization of the method should allow this boundary to be refined. The coverage of the human kinome is approximately 80%, but there remain over 100 kinases for which in vitro inhibition profiles are unknown. The combination of KiPIK screening data with sequence or inhibition-based dendrograms (Fig. 7 and Supplementary Fig. 22) can alleviate this problem, but profiling of additional kinases with existing inhibitor panels will be the best way to increase the coverage of the kinome. The method also requires that the physiological kinase is the most active towards the added substrate in the extract. Lysis of cells means that sub-cellular compartmentation is disrupted, and the use of artificial substrates means that scaffolding and regulatory functions of protein complexes may be impaired. Nevertheless, this criticism also applies to other biochemical approaches and certainly to in silico methods, and it can be

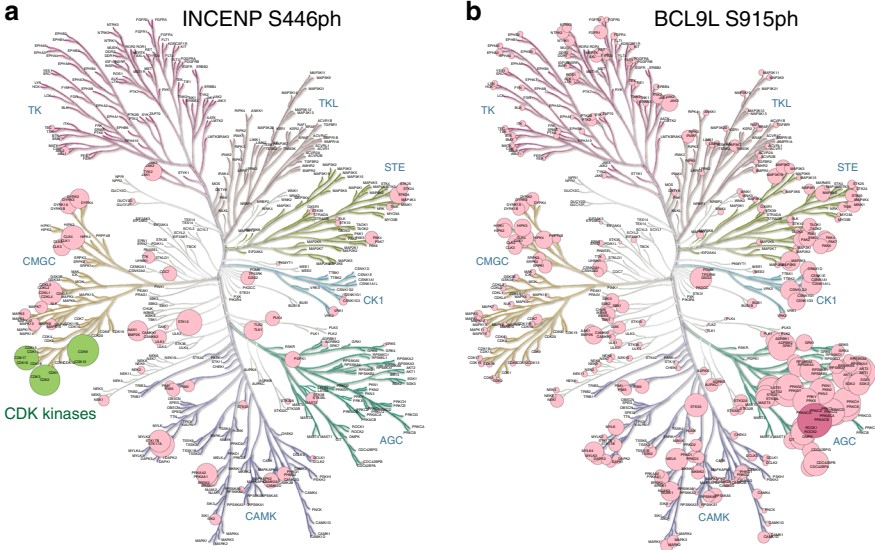

**Fig. 8 KiPIK correlation coefficients for unknown kinases displayed on human kinome dendrograms. a** INCENP S446ph results from the PKIS1/DSF dataset. **b** BCL9L S915ph results from the PKIS2/Repeat 2 dataset. Circle sizes indicate correlation coefficients (from no circle at $\rho = 0$ to the largest circle at $\rho = 1$). For A, strong hits on the CDK kinase branch are shown in green. For B, the top hit is shown in deep violet. Trees for all results can be found in Supplementary Figs. 19 and 20. Kinome trees were produced using KinMap[92], and the underlying tree illustration is reproduced courtesy of Cell Signaling Technology, Inc. (www.cellsignal.com). Source data are provided as a Source Data file.

addressed by using recombinant proteins or protein complexes as substrates in KiPIK, and sub-cellular fractionation could be used to focus screening on particular compartments. We argue that KiPIK is less sensitive to indirect effects than screens using intact cells, but this may vary from case to case. In common with all screening methods, follow up experiments are required to confirm the relevance of candidate kinases. Finally, when more than one kinase phosphorylates the substrate to a significant extent, and these kinases have similar inhibition profiles, KiPIK may be able to reveal candidate kinases that are masked by redundancy in genetic screens (see Fig. 6). However, when kinases with distinct inhibition profiles are involved, then KiPIK in its current form may not always be able to resolve the relevant candidate kinases. Whether such signals can be deconvoluted computationally is an interesting question for further work. Possible approaches have been tried, with varying success, to understand the contributions of multiple kinases to cellular phenotypes using kinase inhibitor screens[77–81].

We can envision several other ways in which KiPIK can be extended and improved in the future. For example, larger inhibitor libraries would allow more robust identification of kinases with little increase in the time required. The use of single-step homogenous kinase assays rather than ELISAs could streamline the procedure, and alternative detection approaches would allow for multiplexing and/or reduce the need for phosphosite-specific antibodies. For example, the use of high-throughput mass spectrometry would allow multiple phosphorylation sites on one or more substrates to be monitored simultaneously without requiring antibodies[28]. It should also be possible to expand the sources of extract used for screening. For example, organoid cultures might provide a more physiological source of cells, and patient-derived material such as blood cells or biopsies might be used to conduct KiPIK screens to obtain insight into disease, such as mechanisms of resistance to therapeutics in cancer. Profiling panels including larger numbers of disease-relevant mutant kinases could also be developed. Given the low proportion of known phosphorylation sites for which kinases are known, and the critical role of kinase signaling in biology and drug development, we expect the method to find wide application.

## Methods

**Cell culture**. HeLa (ATCC CCL 2.2) and A431 (ATCC CCR-1555) cells were grown in DMEM with 5% (v/v) FBS and 100 U/ml penicillin–streptomycin, at 37 °C and 5% $CO_2$ in a humidified incubator.

**Antibodies**. For KiPIK screening, immunoblotting and immunofluorescence microscopy, rabbit polyclonal antibodies to H3T3ph (B8634)[47], INCENP (P240, Cell Signaling Technology #2807), INCENP-S446ph (Jan-Michael Peters, IMP, Vienna)[57], INCENP-TSSph (Michael Lampson, University of Pennsylvania)[82], BCL9L S915ph (Cell Signaling Technology #13325), Neurogranin S36ph (Merck-Millipore, ABN426) and γ-tubulin (AK-15, Sigma, T3320); rabbit monoclonal antibodies to the S-pT-P motif (D73F6; Cell Signaling Technology #5243), Cyclin B1 (D5C10; Cell Signaling Technology #12231) and Vinculin (E1E9V; Cell Signaling Technology #13901); mouse monoclonal antibodies to H3S28ph (CMA315)[83] and phospho-tyrosine (P-Tyr-100; Cell Signaling Technology #9411); and sheep polyclonal antibodies to Aurora B (SAB.1, Stephen Taylor, University of Manchester)[84] and BCL9L (R&D Systems, AF4967) were used. Antibodies used for siRNA screening were mouse monoclonal anti-H3T3ph (16B2)[85] and rabbit polyclonal anti-H2BS6ph[86]. Secondary antibodies were: donkey anti-sheep IgG-HRP (ThermoFisher, A16041), goat anti-rabbit IgG-HRP (Cell Signaling Technology #7074), horse anti-mouse IgG-HRP (Cell Signaling Technology #7076), donkey anti-mouse Alexa Fluor 488, anti-rabbit Alexa Fluor 594 and anti-sheep Alexa 488 Fluor (ThermoFisher, A-21202, A-21207, and A-11015).

**Small molecules and peptides**. KiPIK inhibitor details are provided in Supplementary Data 2–4. Additional compounds used were Forskolin and ZM447439 (Tocris Biosciences), RO-3306 (Calbiochem), H-89 (BioMol International), SU-14813 (Cambridge Biosciences) and MK-2206 (Cayman Chemical).

BCL9L(908–922) peptide: biotin-RGLGRRPSDLTISIN (custom, Eurogentec).

EGFR peptide: biotin-ADEYLIPQQ (AS-29942-1, Eurogentec).

H3(1–21) peptide: ARTKQTARKSTGGKAPRKQLA-GGK-biotin (custom, Abgent).

H3(21–44) peptide: ATKAARKSAPATGGVKKPHRYRPG-GK-biotin (AS-64440, Eurogentec).

INCENP peptide: biotin-GPREPPQSARRKRSY (custom, Eurogentec).

Integrin β1A tail peptide: biotin-GG-KSAVTTVVNPKYEGK (custom, Eurogentec).

Neurogranin(28–43) peptide: biotin-LC-AAKIQASFRGHMARKK (101472-001, Mimotopes).

**Preparation of KiPIK cell extracts**. To prepare mitotic cell extract, HeLa cells were treated with 300 nM nocodazole for 9 h and then collected by shake off. To prepare TPA-stimulated cell extracts, HeLa cells were treated with 200 nM TPA for 30 min. To prepare EGF-stimulated cell extracts, A431 cells were trypsinised, collected in prewarmed DMEM supplemented with 50 ng/ml human EGF (Cell Signalling Technology) and incubated at 37 °C for 5 min. After these treatments, cells were

washed once with PBS and lysed at 4 °C in 50 mM Tris, 0.25 M NaCl, 0.1% Triton X100, 10 mM MgCl₂, 2 mM EDTA, 1 mM DTT, pH 7.5 with protease inhibitor cocktail (Sigma P8340), PhosSTOP (Merck), 1 mM PMSF, 0.1 μM okadaic acid, 10 mM NaF, 20 mM β-glycerophosphate, at approximately $30 \times 10^6$ cells/ml. Extracts were immediately flash frozen in liquid nitrogen.

**KiPIK screens.** Kinase reactions contained 10 μM kinase inhibitor or DMSO (vehicle control), 0.1–0.35 μM peptide, 0.2–0.7 mM ATP, 0.5–5% cell extract, in KiPIK buffer (50 mM Tris, 10 mM MgCl₂, 1 mM EGTA, 10 mM NaF, 20 mM β-glycerophosphate, 1 mM PMSF, pH 7.5 with PhosSTOP (Merck). Reactions were carried out in duplicate in 384-well microplates (ThermoFisher), at 30 °C for 30 min, in a total volume of 35 μl/well.

For detection of phosphorylation, High Capacity Streptavidin-coated 384-well plates (Pierce) were washed thrice with TBS, 0.05% Tween20. In some cases, 10 μl/ well of 500 mM EDTA was added. Then, 35 μl/well of completed KiPIK extract kinase reaction was added and incubated at room temperature for 2 h. After washing, primary phospho-specific antibodies were added at 40 μl/well in TBS, 0.05% Tween20, 0.1% BSA, for 2 h at room temperature (H3S28ph 0.2 μg/ml, H3T3ph 0.1 μg/ml, P-Tyr-100 0.15 μg/ml, INCENP S446ph 1:5000, BCL9L S915ph 1:5000). After washing, 40 μl/well of HRP-conjugated secondary antibodies in TBS, 0.05% Tween20, 0.1% BSA were added for 1 h at room temperature. After washing, HRP-conjugated antibody binding (1:3000) was detected using TMB substrate (New England Biolabs) according to the manufacturer's instructions. All pipetting was performed with a Biomek FX liquid handling robot (Beckman Coulter). Absorbance readings were made using a Polarstar Omega microplate reader (BMG Labtech).

**Analysis of KiPIK results.** Using the mean of duplicate determinations, a standard score was calculated for each inhibitor treatment where standard score = (mean absorbance in presence of inhibitor - mean absorbance of multiple DMSO controls)/standard deviation of DMSO controls. The inhibitory effect of each compound was defined as follows: %inhibition = 100 × (standard score for inhibitor/ lowest standard score on plate). The lowest standard score on the plate was either from the EDTA-treated control wells or from those containing the most inhibitory compound. The %inhibition scores for all inhibitors were then compiled as the inhibition fingerprint of the phosphorylation event probed. Using Prism (GraphPad), Pearson's correlation ($\rho$) was then calculated for this fingerprint against the inhibition profiles of each of the kinases profiled in vitro against that inhibitor library, excluding mutant kinases. Notably, because Davis et al. determined $K_d$ values for each kinase-inhibitor interaction in their panels, rather than single point percent inhibition values[33], we made estimates of the %inhibition expected at 0.5 μM, using the Hill equation: %inhibition = $100/(1 + (K_d/500))$. To calculate z-scores, we used a VBA macro in Excel (Microsoft) to permute inhibitor name labels and generate 100 randomised sets of %inhibition results. From these, we calculated 100 sets of Pearson correlation coefficients to estimate the null distribution of correlation coefficients for that screen (Supplementary Fig. 11A). We then calculated z-scores where z = (observed $\rho$ − mean $\rho$ of null distribution)/ standard deviation of null distribution.

**Inhibitor downsampling and cluster analysis.** To investigate how robust KiPIK results were to the size of the inhibitor library used, we downsampled inhibitor libraries and tested how easy it was to identify true hits using the downsampled versions. From existing libraries of size $N$, we repeatedly sampled (without replacement) $n$ inhibitors and checked how many times the true hit was contained in the top $k$ kinases ranked by correlation with the unknown kinase (Pearson's correlation coefficient, calculated using the $cor$ function in R). By setting $k = 1$, 5 or 10 (or $k \approx N/100$, $N/20$ or $N/10$) and setting $n = 3,...,N$ and sampling 10,000 times, we generated curves representing how the uncertainty in hit identification varies with library size.

For hierarchical clustering of inhibition fingerprints, we used *hclust* and *heatmap* in the *ComplexHeatmap* v1.10.2 package[87] implemented in R version 3.4.0 (2017–04–21)[88], calculating distances using (1 − Pearson correlation coefficient) and clustering using the method Ward.D2[89]. For bootstrapping, we used *hclust* within *pvclust* package 4 with 10,000 iterations, producing two types of p-value: Approximately Unbiased (AU) p-value and Bootstrap Probability (BP) value. The AU value is considered the best indication of robustness[90].

**Kinome siRNA screen.** Hela cells were grown in clear-bottomed microplates (uClear 384 well microplates, Greiner). Transfection was carried out using Interferin-HTS (Polyplus), according to the manufacturer's instructions. The siRNA library targeted 840 genes from the Dharmacon Human Druggable siGENOME library, of which 712 were kinase-related, 110 were G-protein coupled receptor-related and 18 were phosphatase-related. After 24 h, cells were accumulated in mitosis for 6 h by addition of 200 nM nocodazole, immunofluorescently stained with mouse anti-H3T3ph (0.1 μg/ml) and rabbit anti-H2BS6ph antibodies (1:1000), followed by fluorophore-conjugated secondary antibodies, and the intensity of staining quantified by High Content Imaging. Experiments were carried out in quadruplicate. The standard score for condition was then calculated, where standard score = (mean staining intensity in presence of siRNA − mean staining intensity for entire plate)/standard deviation of staining intensity for the entire plate; p-values were calculated using one-sample t-test for each siRNA treatment. Approximately 300 cells were measured for each siRNA treatment.

**Immunofluorescence.** Cells were fixed for 10 min with 2% paraformaldehyde in PBS, washed twice in PBS, then permeabilized for 2 min with 0.5% Triton X100, PBS. After washing twice in PBS, cells were incubated for 1 h in 5% milk in PBS, 0.05% Tween20 at room temperature, and then with primary antibodies at 37 °C in 5% milk in PBS, 0.05% Tween20 (Aurora B 1:2000, INCENP S446ph 1:2000, INCENP TSSph 1:1000). After washing twice with PBS, 0.05% Tween20, and twice with PBS, cells were incubated for 45 min with 2 μg/ml fluorophore-conjugated secondary antibodies at 37 °C in 5% milk in PBS, 0.05% Tween20. After washing twice with PBS, 0.05% Tween20, twice with PBS, and once with milliQ H₂O, samples were mounted using Prolong Gold (Invitrogen) for cover slips, or Fluoromount-G with DAPI (Invitrogen) for microplates.

**Microscopy.** For High Content imaging of cells in microplates, the average fluorescence intensity (integrated) of mitotic cells (defined as cells positive for either stained mitotic histone mark) was measured using a widefield Nikon Eclipse Ti-E microscope equipped with an automated stage, DS-Qi1MC camera (Nikon) and High Content Analysis software (Nikon Elements with JOBS 4.12) using a Plan Apo Lambda 20x/0.8 objective and 1.5x internal magnification. Cells on coverslips were imaged with a Zeiss AxioImager Z2 microscope using a Plan-Apochromat 100x/1.40 oil objective and ZEN 2.3 software (Zeiss). Optical sections were acquired every 0.1 μm using an AxioCam MR R3 camera. Image stacks are displayed as maximum intensity projections.

**Recombinant proteins.** GST-CDK1/GST-CycB kinase was from Cell Signaling Technology (#7518), PKA-α from Millipore (14-440), PKC-δ from Promega (V3401), and Aurora B/INbox(825–918) from Millipore (14-835). Purified recombinant fragments of INCENP were from Novus Biologicals (pPEPTIDE-INCENP(369–583); NBP2-37471CUSTOM) and Upstate Biotechnology (GST-INCENP(825–918); 12-534). Purified full length INCENP-Aurora B complex was produced using the plasmid pGEX-hAurora B-rbs-hINCENP[91] with a 6xHis coding sequence added to the 3′-end. Expression in *E. coli* strain BL21-CodonPlus DE3 (Agilent Technologies) was induced with 1 mM IPTG for 20 h at 18 °C. Bacterial pellets were resuspended in lysis buffer (20 mM Tris-HCl, 0.5 M NaCl, 1 mM DTT, pH 8) with cOmplete™ Mini EDTA-free Protease Inhibitors (Merck). Cells were lysed by sonication, polyethylenimine was added to precipitate cellular debris and, after centrifugation twice at 3600 g for 5 min, the supernatant was incubated with glutathione Sepharose 4B resin (GE Life Sciences) for 15 min. After washing with 5 volumes of lysis buffer, the resin was incubated for 10 min in lysis buffer containing 5 mM ATP, 5 mM MgCl₂ to eliminate bacterial chaperones. Recombinant proteins were then eluted with 150 mM Tris-HCl, 0.5 M NaCl, 1 mM DTT, 50 mM reduced glutathione, pH 9.2 and dialysed into 50 mM Tris-HCl, 0.3 M NaCl, 0.5 mM EDTA, 1 mM DTT, pH 8 overnight at 4 °C, in the presence of PreScission protease (Merck) to cleave GST. Following a second dialysis into 20 mM NaPO₄, 75 mM NaCl, pH 7 at 4 °C for 4 h, and removal of precipitates by centrifugation at $3600 \times g$ for 5 min, free GST was removed by adding glutathione Sepharose 4B. Recombinant proteins were further purified by ion exchange chromatography (Q Sepharose; GE Life Sciences), elution with increasing concentration of NaCl, and then dialysis into 20 mM NaPO₄, 150 mM NaCl, 1 mM DTT, pH 7. Finally, purified INCENP-Aurora B was treated with λ phosphatase (New England Biolabs) to remove existing phosphorylation.

**In vitro kinase reactions.** When using INCENP fragments as substrates, reactions were carried out in 20 mM HEPES, 1 mM ATP, 5 mM MgCl₂, 0.14 M NaCl, 3 mM KCl, pH 7.4 with ~100 nM of each substrate and 10 nM of recombinant kinase for 30 min at 30 °C. When using full length INCENP-Aurora B substrate, reactions were carried out in 50 mM Tris, 1 mM ATP, 10 mM MgCl₂, 1 mM EGTA, 10 mM NaF, 20 mM β-glycerophosphate, pH 7.5 with PhosSTOP phosphatase inhibitors (Merck) and approximately 70 nM of INCENP substrate and 7 nM of recombinant Cyclin B1-Cdk1 for 30 min at 37 °C. Aurora B inhibitor ZM447439 was included at 0.5 μM. To assay full-length BCL9L phosphorylation, HeLa cells lysed in 50 mM Tris, 0.15 M NaCl, 0.1% SDS, 1% Triton X-100, 0.5% sodium deoxycholate, pH 7.5, with 2.5 U/μl Benzonase (Merck) and cOmplete™ Protease Inhibitor Cocktail (Roche) were subjected to immunoprecipitation with 5 μg/ml anti-BCL9L antibodies and Dynabeads Protein G (ThermoFisher). After 1 h, beads were washed three times in PBS, split into 3 equal fractions and incubated with 1 nM PKA-α or PKC-δ, or without kinase, at 1 mM ATP in the conditions described above for peptides. After washing, bound proteins were eluted by boiling in SDS-PAGE sample buffer and analyzed by immunoblotting. When using peptide substrates, reactions were carried out at 37 °C for 30 min with 0.1 μM peptide and 0.1 mM ATP and either PKA-α in 50 mM Tris, 10 mM MgCl₂, 1 mM EGTA, 10 mM NaF and 20 mM β-glycerolphosphate, pH 7.5, or with PKC-δ in 40 mM Tris, 20 mM MgCl₂, 0.1 mg/ml BSA, 50 μM DTT, 50 μg/ml phosphatidylserine, 5 μg/ml 1-stearoyl-2-linoleoyl-sn-glycerol, 5 μg/ml 1-oleoyl-2-acetyl-sn-glycerol, pH 7.5, and phosphorylation was detected using antibodies as described for KiPIK screens (BCL9L S915ph 1:1000, Neurogranin S36ph 1:1000).

**INCENP immunoprecipitation**. HeLa 1C8 cells[47] were treated with 0.5 µg/ml nocodazole for 8 h. After collecting mitotic cells by shake off and replating, either 100 µM Roscovitine or DMSO (vehicle control) was added together with 20 µM MG132 in the continued presence of nocodazole. After 2 h, cell lysates were prepared in 50 mM Tris, 0.3 M NaCl, 0.5% Triton X-100, 10 mM MgCl₂, 5 mM EDTA, 1 mM dithiothreitol, 1 mM PMSF, 0.1 µM okadaic acid, 10 mM NaF, 20 mM β-glycerophosphate, 0.25 U/µl Benzonase, cOmplete™ Mini EDTA-free Protease Inhibitors (Merck), pH7.4 and subjected to immunoprecipitation with rabbit anti-INCENP (diluted 1:30) and Dynabeads Protein G (ThermoFisher). After washing, bound proteins were eluted by boiling in SDS-PAGE sample buffer.

**RNAi and inhibitor treatment of cells**. HeLa cells were transfected with 50 nM Mission siRNAs using Lipofectamine RNAiMAX according to the manufacturer's instructions (ThermoFisher). After 48 h, cells were treated with 200 nM TPA for 30 min before lysis in NuPAGE loading buffer (ThermoFisher) containing 100 mM DTT, 0.25 U/µl Benzonase, and PhosSTOP (Merck). The siRNAs were: BCL9L, EH054121; AKT1, EHU083501; AKT3, EHU067201; PKA-α, EHU132541; PKA-β, EHU075621; PKC-δ, EHU067081; PKC-ε, SIHK1828; PKC-η, SIHK1833; PKG1 #1, SIHK1862; PKG1 #5, SIHK1866; PKG2 #1, SIHK1867; PKG2 #2, SIHK1868; LATS2, SIHK1044 (Sigma-Aldrich). Alternatively, cells were treated with 200 nM TPA or 10 µM Forskolin for various time periods, or were pretreated with 0.5 µM SU-14813, 0.2 µM Gö 6983, 30 µM H-89 or 10 µM MK-2206 for 30 min and then stimulated with 200 nM TPA for 30 min before lysis in 50 mM Tris, 0.15 M NaCl, 0.1% SDS, 1% Triton X-100, 0.5% sodium deoxycholate, pH 7.5, with 2.5 U/µl Benzonase (Merck) and cOmplete™ Protease Inhibitor Cocktail (Roche).

**SDS-PAGE and immunoblotting**. SDS-PAGE was carried out using the NuPAGE Bis-Tris (ThermoFisher) or BioRad systems. Proteins were transferred onto PVDF or Nitrocellulose membranes using standard techniques. Membranes were blocked with 3–5% milk in TBS, 0.1% Tween20 for 1 h, then with primary antibodies in 3–5% milk, TBS, 0.1% Tween20 overnight at 4 °C (INCENP 1:1000, INCENP S446ph 1:1000, INCENP TSSph 1:1000, BCL9L 0.4 µg/ml, BCL9L S915ph 1:1000, Vinculin 1:1000, Cyclin B1 1:1000, S-pT-P 1:1000, γ-tubulin 8 µg/ml). After washing 3 times with TBS, 0.1% Tween20, they were incubated for 1 h with HRP-conjugated secondary antibodies (1:10,000) in 3–5% milk, TBS, 0.1% Tween20, washed three times with TBS, 0.1% Tween20 and once with TBS, before development using ECL substrate (Pierce or GE Healthcare) and exposure to X-ray film. Stripping of immunoblots was carried out with Restore Western Blot Stripping Buffer (ThermoFisher).

**Reporting summary**. Further information on research design is available in the Nature Research Reporting Summary linked to this article.

## Data availability

All relevant data are available from the authors. The source data underlying Figs. 2, 3, 4a–c, 5–8 and Supplementary Figs. 2–8 and 11–20 are provided as a Source Data file. Data for Supplementary Figs. 9, 10, and 21–32 can also be found at https://github.com/CnrLwlss/Watson_2020.

## Code availability

R source code can be found at https://github.com/CnrLwlss/Watson_2020 and the VBA Shuffler code is provided in the Source Data file.

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

## Acknowledgements

We would like to thank the late Robert Stones for carrying out clustering analysis in the early phases of this work, Peter Banks and Alex Laude for their invaluable help with screening robotics and bioimaging, respectively, Bethany Weston for help with RNAi experiments, Daniel Rico for access to computing facilities, and Julian Higgins for discussions of statistics. PKIS libraries were supplied by GlaxoSmithKline LLC and the Structural Genomics Consortium under an open access Material Transfer and Trust Agreement: http://www.sgc-unc.org. This study was funded by a Wellcome Trust Investigator Award and a Royal Society Wolfson Research Merit Award to JMGH, an MRC DiMeN PhD studentship to MCD and by JSPS KAKENHI JP17H01417 and JP18H05527HK to HK.

## Author contributions

N.A.W., T.N.C., and O.S. carried out experiments; C.L., M.C.D., T.N.C., and J.M.G.H. conducted computational analyses; N.A.W., T.N.C., C.L., and J.M.G.H. designed experiments and analysis approaches; N.A.W., T.N.C., C.L., M.C.D., and J.M.G.H. interpreted data; K.S. and T.H. made recombinant proteins; H.K. provided antibodies; N.A.W. and J.M.G.H. conceived the project and wrote the majority of the paper. All authors commented on and contributed to the paper.

## Competing interests

Nikolaus Watson and Jonathan Higgins are the inventors on UK patent filing No. 1906445.0 "Kinase screening assays" that covers the procedure described in this manuscript. The other authors declare no competing interests.
