## [Peer Review File · Nature Communications]

Reviewers' comments:

Reviewer #1 (Remarks to the Author):

Thank you for this submission. I enjoyed reading this clear, well written paper. You describe the development of KiPIK, methodology that can be used when one has a residue of interest that gets phosphorylated and want to know "What kinase (kinases) does that?". This is complementary to the approaches utilized when one has a kinase of interest and want to know what substrates it can phosphorylate. You provide ample background on methodologies used by others, providing pros and cons for these methods and your own. I appreciate your recognition of the shortcomings of each method and the follow up experiments required in all cases. This to me is an even handed reporting of the lay of the land for this field. Your method nicely takes advantage of kinase inhibitor profiling data in the literature and uses it in conjunction with inhibition of phosphorylation events in cells. I am convinced others will apply this strategy and expand on it with other inhibitor sets that provide further coverage, and with different methods to monitor phosphorylation (such as mass spec) as you have described.

I have a few minor suggestions.

In the section "The KiPIK method" you have a sentence that starts "Suitable libraries include...". I suggest modifying the descriptions of the libraries slightly to better represent their origins. For example "Suitable libraries include the Published Kinase Inhibitor Set 1 (PKIS1) which consists of published GlaxoSmithKline Kinase inhibitors (reference 34), a set of diverse kinase inhibitors compiled by scientists at the Fox Chase Cancer Center (reference 31, and a set of literature kinase inhibitors assembled by a team at EMD Millipore (reference 32)." It may also help the reader to have the number of compounds screened in each set here, along with the different assay formats used, but you could choose to have that detailed information later in the manuscript.

In the "KiPIK screens for tyrosine phosphorylation sites identify the expected kinases" section:

You say "biotinylated substrate peptides" I could not find the specific substrates you used in this case. If I missed it I apologize. Will you please include this important detail for the readers?

In the "Inhibitor library requirements for KiPIK screening" section:

In one paragraph you say you assembled a custom library. Would you please explain how/why you chose the compounds to include in this custom library? Why did you choose the 76 out of the 178 and the 55 out of the 158?

For my curiosity rather than changes / additions, but perhaps useful to others: You don't talk much about the values of the Pearson's coefficients. Do you simply use it to guide you to the best scorers and then follow up / validate as appropriate?

Have you explored dosing at multiple concentrations and looked at dose responsive correlations? 10 micromolar is high for some of these inhibitors and of course higher concentration brings more off targets along.

Reviewer #2 (Remarks to the Author):

In this manuscript Watson et al present an approach to identify kinases that phosphorylate specific phosphorylation sites on proteins. The method consists of assessing the effect of kinase inhibitors on the activity of kinases present in cell extracts. This is achieved by incubating cell lysates with a short peptide that is suspected to be kinase substrate and a panel of kinase inhibitors. The effects of the inhibitors are then correlated with the known selectivity of such compounds towards kinases. A simple correlation coefficient is then used to rank kinases and thus predict the kinase that phosphorylates the assayed substrate. The authors show that the approach works for four different kinase substrates. However, the general applicability of the method is questionable because the potential for false discovery is very high and this diminishes enthusiasm for the study. Furthermore, there are several issues with how the information is presented that will limit the ability of other labs to reproduce the results of the study.

1. The list of kinase inhibitors and their kinase targets is not given. Supplementary dataset 2 lists the kinases covered by different published screens but not the compounds that inhibit such kinases or the potency of inhibition. Without these data, the information in Supplementary dataset 2 is worthless, and the results of the experiments will not be reproducible by other laboratories.

2. It is not clear how the authors decide that a kinase is responsible for phosphorylating a given peptide substrate. The ranking of correlation coefficients produces ambiguous results. For example, in the case of phospho-H3S28, the correlation coefficient for Aurora B is 0.62 or 0.74; integrin b1a Y795 correlation with FGR is 0.5; and for EGFR with Y1016 the correlation is ~ 0.55 (Figure 2). Using the same criteria, GSK3A, GSK3B and STK16 should be kinases that phosphorylate INCENP at S446 because the correlation for these kinases and this site are >0.5 (Figure 4). However, the authors claim that it is CDK1-cyclinB the kinase responsible for phosphorylating this residue. Thus comparing the results shown in Figures 2 and 4 highlights the ambiguity of the results produced by this method.

3. The study is based on assaying the phosphorylation of peptides by endogenous kinases in cell lysates in-vitro. However, the substrate specificity of some kinases depends on long range interactions as well as recognition of motifs surrounding the site of modification. The data suggest that is phosphorylated by CDK1, but this is only demonstrated in vitro using a fragment of INCENP and not the full length protein. Can the authors rule out that GSK3A or GSK3B phosphorylate INCENP at pS446?

4. In the manuscript only one new putative kinase-substrate relationship is presented and the results are not convincing. The authors should show that the method allows identifying new substrates for more one kinase.

5. For the method to work the kinase inhibitor fingerprint for a given kinase needs to be different from those of other kinases. Do unique fingerprints exist for all kinases? For example, can the method distinguish substrates that are specific for related kinases, such AKT1 and not for AKT2 or for ERK1 and not for ERK2? Or for a less challenging case such as AKT vs S6K?

6. Another issue is that the information on kinase inhibitor specificity is incomplete. This lack of comprehensive information could in principle lead to false discovery in cases when the actual inhibitor fingerprint for a given kinase is not reflected in the available information present on the inhibitor selectivity screens.

7. Another source of false positives is that cell extracts are used for the approach but, although the authors claim that this is an advantage, the problem with this is that the approach measures kinase activity outside the context of their intracellular location.

8. The authors exaggerate the simplicity of the method. For example, in the discussion section the authors write that “we describe a new simple and rapid method...”. However, assaying >100 kinase inhibitors is not simple for most labs and it is only “simple and rapid” after complex and time consuming optimization of automation procedures.

9. The authors should check that the kinase that is being predicted is actually expressed in the cell line that is being assayed.

10. It is not clear why a 10 micromolar inhibitor concentration was chosen for the assays given that most of these compounds inhibit kinases in the nanomolar range.

11. The vendor or source of the compounds in the inhibitor libraries used for the screen should be given.

Reviewer #3 (Remarks to the Author):

Review of “KiPIK, a new approach to identify kinases for specific phosphorylation sites”

In the manuscript, Dr. Higgins and coworkers developed a new technology to identify the specific kinase that phosphorylates any given phosphorylation site that they termed “KiPIK”. Using large-scale kinase inhibitor libraries paired with pre-existing in vitro kinase inhibitor response databases, they are able to generate inhibition profiles providing a “fingerprint” that allows unknown kinases that phosphorylate a specific phospho-site in cell lysates to be determined. Furthermore, they validate KiPIK’s effectiveness to identify unknown kinases in cell lysates using 2 phospho-sites with established kinases (Histone H3 and EGFR Y1068) and 1 phospho-site in INCENP with no known kinase. The authors then establish that the commonly phosphorylated motif in INCENP was a non-conical target of Cyclin B-Cdk1. Finally; the authors show that KiPIK has advantages over pre-existing methods for assigning kinases to phospho-sites such as RNA interference strategies by eliminating

detection of indirect effects observed by inhibition of upstream kinases. As the current methods for assigning kinase phosphosite dependencies are limited, the ability to rapidly identify kinases using KiPIK would be tremendously advantageous, particularly in understanding kinase biology and development of resistance to kinase inhibitors.

Major concern:

Although the authors elegantly demonstrate that KiPIK can identify a previously unknown kinase that phosphorylates INCENP, further demonstration of the utility of KiPIK to identify unknown kinases will be required before the manuscript should be considered for publication. In order for the technology to be termed “generally applicable” at least 3 more examples of KiPIK need to be performed, including phosphorylation sites with no consensus motifs.

Taken together, the manuscript is well written and provides a potentially valuable technology for assigning unknown kinases for specific phosphorylation sites, however the manuscript is too preliminary in its current form.

Reviewers' comments:

Reviewer #1 (Remarks to the Author):

Thank you for this submission. I enjoyed reading this clear, well written paper. You describe the development of KiPIK, methodology that can be used when one has a residue of interest that gets phosphorylated and want to know "What kinase (kinases) does that?". This is complementary to the approaches utilized when one has a kinase of interest and want to know what substrates it can phosphorylate. You provide ample background on methodologies used by others, providing pros and cons for these methods and your own. I appreciate your recognition of the shortcomings of each method and the follow up experiments required in all cases. This to me is an even handed reporting of the lay of the land for this field. Your method nicely takes advantage of kinase inhibitor profiling data in the literature and uses it in conjunction with inhibition of phosphorylation events in cells. I am convinced others will apply this strategy and expand on it with other inhibitor sets that provide further coverage, and with different methods to monitor phosphorylation (such as mass spec) as you have described.

I have a few minor suggestions.

1. In the section "The KiPIK method" you have a sentence that starts "Suitable libraries include...". I suggest modifying the descriptions of the libraries slightly to better represent their origins. For example "Suitable libraries include the Published Kinase Inhibitor Set 1 (PKIS1) which consists of published GlaxoSmithKline Kinase inhibitors (reference 34), a set of diverse kinase inhibitors compiled by scientists at the Fox Chase Cancer Center (reference 31), and a set of literature kinase inhibitors assembled by a team at EMD Millipore (reference 32)." It may also help the reader to have the number of compounds screened in each set here, along with the different assay formats used, but you could choose to have that detailed information later in the manuscript.

As requested, we have modified the text to more clearly explain the origin of the libraries. We think detailed information on the inhibitors used is more appropriately described later in the paper because the exact number of compounds used in KiPIK differs somewhat from the published papers, and different combinations are likely to be compatible with the method.

In the "KiPIK screens for tyrosine phosphorylation sites identify the expected kinases" section:

2. You say "biotinylated substrate peptides" I could not find the specific substrates you used in this case. If I missed it I apologize. Will you please include this important detail for the readers?

This information was in the Methods under "Peptides" (now "Inhibitors and Peptides")

3. In the "Inhibitor library requirements for KiPIK screening" section:

In one paragraph you say you assembled a custom library. Would you please explain how/why you chose the compounds to include in this custom library? Why did you choose the 76 out of the 178 and the 55 out of the 158?

Davis *et al.* profiled 72 inhibitors, Anastassiadis *et al.* profiled 178 inhibitors, and Gao *et al.* 158 inhibitors. We selected all 72 of the compounds used to generate the most extensive kinome profile among these relevant papers (ie Davis *et al.*). We supplemented this with all compounds available from Selleck that were present in the EMD Millipore compound set used in both Gao *et al.* and Anastassiadis *et al.* studies. Finally, we included 4 additional compounds that the Fox Chase group added to the EMD Millipore set (Anastassiadis *et al.*) that were not present in the Davis *et al.* set. This resulted a library of 128 inhibitors. There was no systematic selection of inhibitors based on their properties. We have added an explanatory note to Supplemental Table 3 where the custom library is described.

4. For my curiosity rather than changes / additions, but perhaps useful to others: You don't talk much about the values of the Pearson's coefficients. Do you simply use it to guide you to the best scorers and then follow up / validate as appropriate?

This is an interesting question, also brought up by reviewer 2, and we have added discussion of it on pages 10 and 14. As the reviewer implies here, we consider the method to be a ranking system. It tells a user in what order to test kinases in follow-up assays. We would argue that "competitor" methods are effectively similar in this respect. For example, an RNAi screen in cells will always yield an siRNA that produces the strongest effect. In one screen, this might be 90% inhibition, and in another it might be 60%. However, investigators are likely to recognise that the effect of kinase knockdown will vary for different substrates and for different readouts. It is, however, reasonable to ask at what point a KiPIK correlation is too weak to be meaningful. We do not have a rigorous answer to this question but we have now introduced a new metric, the z-score, which enumerates how far away from the centre of the expected null distribution a given correlation score is, and provides an indication of the robustness of the results. Empirically, we would consider a screen where all of the z-scores are below 6 to be questionable. In addition, inspection of the correlation plots (see Supplemental Figures 2 to 6, 12) can provide an idea whether lower correlation scores are driven by multiple inhibitors, or one or two outliers. These inhibitors

can then be tested in cell assays to guide kinase prioritisation. Also please see the response to Reviewer 2, point 2.

5. Have you explored dosing at multiple concentrations and looked at dose responsive correlations? 10 micromolar is high for some of these inhibitors and of course higher concentration brings more off targets along.

The principle upon which KiPIK is based is that we do not make any distinction between “on target” and “off target” inhibition. We simply make use of the entire inhibition profile, whatever it is. It is true, however, that the inhibitor concentration used in the extract KiPIK assays should “match” the concentration used in the acquisition of recombinant kinase inhibitor profiling data. In this case, a “match” does not necessarily mean the same concentration of inhibitor should be used in both assays: it is more important that the %inhibitions of a given kinase in the KiPIK assay should be similar to those in the profiling data, so that correlation coefficients can be accurately determined. Note that more inhibitor may be needed in extract assays than in assays with purified proteins due to the presence, for example, of additional inhibitor binding proteins and higher ATP concentrations. Although we have not explored the effect of different concentrations within our extract assays, we have investigated the impact of comparing KiPIK screens carried out at 10 μ M with kinase profiling datasets carried out at different inhibitor concentrations, which provides similar insights. As shown in Supplemental Figure 8, and discussed in the text, mismatches in inhibitor concentrations used to generate KiPIK and profiling datasets can lead to non-linear correlation plots which will lower the ability to distinguish the correlation coefficients for different kinases. This can be monitored by observing the correlation graphs for the highest-ranking kinases.

Reviewer #2 (Remarks to the Author):

In this manuscript Watson et al present an approach to identify kinases that phosphorylate specific phosphorylation sites on proteins. The method consists of assessing the effect of kinase inhibitors on the activity of kinases present in cell extracts. This is achieved by incubating cell lysates with a short peptide that is suspected to be kinase substrate and a panel of kinase inhibitors. The effects of the inhibitors are then correlated with the known selectivity of such compounds towards kinases. A simple correlation coefficient is then used to rank kinases and thus predict the kinase that phosphorylates the assayed substrate. The authors show that the approach works for four different kinase substrates. However, the general applicability of the method is questionable because the potential for false discovery is very high and this diminishes enthusiasm for the study. Furthermore, there are several issues with how the information is presented that will limit the ability of other labs to reproduce the results of the study.

1. The list of kinase inhibitors and their kinase targets is not given. Supplementary dataset 2 lists the kinases covered by different published screens but not the compounds that inhibit such kinases or the potency of inhibition. Without these data, the information in Supplementary dataset 2 is worthless, and the results of the experiments will not be reproducible by other laboratories.

The inhibition profile datasets that the reviewer refers to are published (Elkins *et al.*, Drewery *et al.*, Davis *et al.*, Anastassiadis *et al.*, Gao *et al.*), and it did not seem to us appropriate to republish the data of others. The list of “custom” library inhibitors we used was provided in Table 2 (now Table 3) of the supplemental dataset. For completeness, we have now also added the list of the 312 out of the 367 compounds in the PKIS1 dataset and the 485 out of the 645 compounds in the PKIS2 dataset that we used in our assays (new Table 2 and Table 4). These compounds constitute the entire array of PKIS1 and PKIS2 compounds that we received from the SGC. In addition, complete inhibition profiling data are now included in the “Source Data” files.

2. It is not clear how the authors decide that a kinase is responsible for phosphorylating a given peptide substrate. The ranking of correlation coefficients produces ambiguous results. For example, in the case of phospho-H3S28, the correlation coefficient for Aurora B is 0.62 or 0.74; integrin b1a Y795 correlation with FGR is 0.5; and for EGFR with Y1016 the correlation is ~ 0.55 (Figure 2). Using the same criteria, GSK3A, GSK3B and STK16 should be kinases that phosphorylate INCENP at S446 because the correlation for these kinases and this site are >0.5 (Figure 4). However, the authors claim that it is CDK1-cylinB the kinase responsible for phosphorylating this residue. Thus comparing the results shown in Figures 2 and 4 highlights the ambiguity of the results produced by this method.

We would argue that the results are not so ambiguous. For a given screen, KiPIK produces an unambiguous list of candidate kinases ranked in order of likelihood. This gives investigators clear guidance as to which kinases to follow up in secondary assays. We have not compared the absolute correlation scores between different KiPIK assays, as these are likely to be influenced by the particular biology of each kinase in cell extracts (eg the presence of competing kinases and any indirect effects). This is not an unusual practice. As discussed in the reply to reviewer 1 point 4, for example, the absolute %changes measured for phenotypes in different genetic screens will vary widely from screen to screen, but this does not make them uninterpretable. Researchers would be unlikely to dismiss a top hit just because it has an effect lower than in a different genetic screen, or to prioritise low ranking hits in a screen over the top hits. We agree that the significance of the absolute value of the correlation scores is not fully clear, but we think, in practice, the question becomes “if the highest-ranking

kinase(s) in my KiPIK screen have low correlation scores, how do I interpret the results?" (Analogous to the question, "if the %inhibition of the top hit in my RNAi screen is low, do I ignore the result?"). To address this, we have now introduced a new metric, the z-score, that enumerates how far away from the centre of the expected null distribution a given correlation score is, providing an indication of the robustness of the results. Empirically, we would consider a screen where all of the z-scores are below 6 to be questionable but (as for genetic screens) we do not have a robust way to decide this, as also discussed in point 4 above. We have now added discussion of this point on pages 10 and 14.

3. The study is based on assaying the phosphorylation of peptides by endogenous kinases in cell lysates in-vitro. However, the substrate specificity of some kinases depends on long range interactions as well as recognition of motifs surrounding the site of modification. The data suggest that is phosphorylated by CDK1, but this is only demonstrated in vitro using a fragment of INCENP and not the full length protein. Can the authors rule out that GSK3A or GSK3B phosphorylate INCENP at pS446?

It is true that we have used peptide substrates to develop KiPIK, and this possible limitation is explicitly mentioned in the Discussion where we have tried to provide a balanced view of the strengths and weaknesses of KiPIK. In particular, we pointed out that it is possible to carry out KiPIK screens using full-length substrates, or even substrates within protein complexes. We believe that essentially all methods to find kinases for particular substrates have pros and cons. Our aim here is to describe a method that is useful and that compares favourably with other approaches, not to claim that it is ideal in every respect. It is worth pointing out that widely used computational methods also use information only from sequences immediately surrounding the phosphosite, as do methods using crosslinkable peptide substrates.

As explained above, we consider KiPIK a ranking system, and do not think that our results suggest that GSK3 kinases play a role in INCENP S446 phosphorylation. Indeed, ranking the hits in this screen by z-score gives a top ten list entirely occupied by Cdk kinases, with Cdk1-Cyclin B at number 1 (see Supplemental Figure 11C), confirming that GSK3 kinases are lower confidence hits. However, to address the question raised by the reviewer, we carried out two experiments. First, we compared the ability of GSK3 and Cdk1 inhibitors to reduce phosphorylation of INCENP S446 in mitotic cells, and found that only the Cdk1 inhibitor was active (see figure below). Second, we tested if Cyclin B1-Cdk1 was able to phosphorylate full-length recombinant INCENP, and found that this was the case (Supplemental Figure 7A). The results support the conclusion that Cdk1 and not GSK3 kinases are responsible for this phosphorylation event.

Inhibition of Cdk1 but not GSK3 reduces INCENP S446 phosphorylation in mitosis.

HeLa cells were arrested in mitosis for 12 h in 300 nM nocodazole ("mitosis") or left untreated ("interphase"), then exposed to 10 μM GSK3α/β inhibitor CHIR99021 or Cdk1 inhibitor RO3306 for 5 min or 30 min. After 5 min, Cdk1 inhibition reduces INCENP S446ph, without reducing the mitotic marker H3S10ph, while GSK3 inhibition has no effect. After 30 min, the Cdk1 inhibitor eliminates detectable INCENP S446ph, but by this time the cells are being driven out of mitosis, as reported by the decline in H3S10ph. Again GSK3 inhibition has no effect.

4. In the manuscript only one new putative kinase-substrate relationship is presented and the results are not convincing. The authors should show that the method allows identifying new substrates for more one kinase.

As explained above, we believe that KiPIK unambiguously ranks cyclin-dependent kinases as most likely responsible for INCENP S446ph, and we have further enhanced our validation of this *in vitro* and in cells as requested by the reviewer. We think this validation now exceeds that carried out for a number of other published INCENP phosphorylation sites. The reviewer does not explain his/her logic for requesting validation using more than one new substrate. We believe that, to validate a new screening method, by far the best and most logical approach is to use kinase-substrate relationships that are widely accepted and well established in the literature. We show four such examples. The problem with using new kinase-substrate pairs to validate the method is that the validation can only be as good as the limited number of follow-up assays that our lab can carry out on each one, making it an inferior approach.

Nevertheless, we appreciate that additional examples can only boost confidence in the method, and we have carried out an additional screen to identify the kinase for another previously unassigned phosphosite. In two separate screens, PKA was the top ranked kinase for phosphorylation of BCL9L at S915 following cell stimulation

with phorbol ester. Follow up experiments confirmed that PKA (but not phorbol-ester sensitive PKC) phosphorylated BCL9L peptides and full-length protein, that stimulation of cAMP production by Forskolin boosts BCL9L S915ph, and that PKA RNAi and small molecule inhibitors prevented BCL9L S915ph in cells (Figure 6).

5. For the method to work the kinase inhibitor fingerprint for a given kinase needs to be different from those of other kinases. Do unique fingerprints exist for all kinases? For example, can the method distinguish substrates that are specific for related kinases, such as AKT1 and not for AKT2 or for ERK1 and not for ERK2? Or for a less challenging case such as AKT vs S6K?

This is a good question, and one that we have already addressed experimentally using two families of kinases: Aurora kinases and the EGFR family of kinases. For both the Aurora and EGFR families, our example KiPIK screens find that the correct family members (Aurora B and EGFR) are the top overall hits. This is suggestive that the method can distinguish between very closely related kinases. However, we do not have the power to be able to quantify this precisely and it may be different for different kinase families, as already explained in the Discussion. We would argue, though, that if one carries out a KiPIK screen with no idea about the kinase responsible, and the method unambiguously reveals a particular family of kinases to be the top candidate, this is a huge benefit. In the case of the Aurora and EGFR families, one now has to conduct follow up experiments with three possible kinases (Aurora A/B/C or EGFR/ERRB2/ERBB4) rather than up to 540 kinases in the full kinome. It is worth pointing out that widely used computational methods also have great difficulty in distinguishing closely related kinases, and that genetic screens may fail to identify the role of some kinase families because of redundancy. Indeed, KiPIK identified PKA-alpha/beta as BCL9L kinases, and this was substantiated by double knockdown of PKA-alpha/beta in cells. However, single knockdown of either of these kinases did not noticeably affect BCL9L S915ph, suggesting that KiPIK screening actually may have advantages over genetic screening in this context (see Figure 6).

(NB the results for the Src-family kinases are more difficult to evaluate, because the specific Src kinase(s) carrying out integrin beta1 phosphorylation in A431 cells have not been confirmed, and many Src kinases have similar substrates *in vivo*).

6. Another issue is that the information on kinase inhibitor specificity is incomplete. This lack of comprehensive information could in principle lead to false discovery in cases when the actual inhibitor fingerprint for a given kinase is not reflected in the available information present on the inhibitor selectivity screens.

The profiling information for the kinases included in the datasets used is quite complete, with few missing datapoints (Elkins *et al.*, Drewery *et al.*, Davis *et al.*, Anastassiadis *et al.*, Gao *et al.*). However, it is true that not all kinases are represented in these profiling datasets. This is acknowledged and discussed at some length in the manuscript. We provide two methods to evaluate whether missing kinases may be candidates, most notably plotting the results on kinome trees as shown in Figure 7 and multiple Supplemental Figures. Our aim is not to prove that KiPIK is “all things to all kinases”, but that it is useful and compares favourably to other methods. It is particularly notable that most computational methods are restricted to many fewer kinases than KiPIK, and those that claim high coverage (such as PhosphoNET) are poorly validated (see legend to Supplemental Figure 33). Even methods that appear to cover all kinases (such as RNAi screens) almost certainly do not, as not all kinases will be efficiently depleted (due to factors such as inefficient RNAi sequences, the existence of untargeted kinase splice variants, redundancy, and slow turnover of some kinase proteins) and false discovery is possible due to indirect effects.

7. Another source of false positives is that cell extracts are used for the approach but, although the authors claim that this is an advantage, the problem with this is that the approach measures kinase activity outside the context of their intracellular location.

This is certainly true and, again, it was explicitly acknowledged in the Discussion (“Lysis of cells means that sub-cellular compartmentation is disrupted”). We think all approaches available for identifying kinases have advantages and disadvantages, and we have tried to provide a balanced view of these. The most widely used approach currently (*in silico* prediction using consensus or optimal kinase substrate motifs) essentially excludes all biological context, and so we believe that KiPIK compares well with this. Methods such as kinase purification or substrate cross-linking in extracts suffer the same concerns. Methods that preserve biological context (eg genetic approaches in cells) have their own problems; most notably the higher likelihood of indirect effects.

8. The authors exaggerate the simplicity of the method. For example, in the discussion section the authors write that “we describe a new simple and rapid method...”. However, assaying >100 kinase inhibitors is not simple for most labs and it is only “simple and rapid” after complex and time consuming optimization of automation procedures.

We believe that the method is simple and rapid compared to other experimental approaches to the problem (such as kinome-wide cell screens, or purification from cells by crosslinking to a bespoke substrate or tracking kinase activity followed by mass spectrometry). Nevertheless, we agree that the impression given by this wording may be wrong and we have removed “simple and rapid” from the text.

9. The authors should check that the kinase that is being predicted is actually expressed in the cell line that is being assayed.

Because of the particular kinases involved here, there is extensive literature to support their expression in the cell lines used. A431 cells are well-known to express EGFR and Src kinases. PKA- α/β is ubiquitous, and Aurora B and Cyclin B1-Cdk1 are expressed in essentially all mitotic cells, and certainly in HeLa cells that have been used in hundreds of published studies of these kinases. We (and others) have worked on Haspin (and Aurora B and Cyclin B1-Cdk1) in HeLa cells for many years. Many of these papers are cited in the manuscript. For EGFR and Src in A431 cells see Walton *et al.* 1990, Rotin *et al.* 1992, Oshero *et al.* 1994, Sato *et al.* 1995a, Sato *et al.* 1995b. For Aurora B, Cyclin B1-Cdk1 and Haspin in HeLa cells see Goto *et al.* 2002; Dai *et al.* 2005, Goto *et al.* 2006, Skoufias *et al.* 2007, Dephoure *et al.* 2008, Deibler *et al.* 2010, Wang *et al.* 2011, Hegemann *et al.* 2011, de Antoni *et al.* 2012, Ghenoiu *et al.* 2013, Zhou *et al.* 2014, Wheelock *et al.* 2017.

10. It is not clear why a 10 micromolar inhibitor concentration was chosen for the assays given that most of these compounds inhibit kinases in the nanomolar range.

In a nutshell, this is because the %inhibition data in cell extracts must be generated in a manner that matches the %inhibition data already generated by others for purified kinases *in vitro*. For more detail, please see response to reviewer 1, point 5.

11. The vendor or source of the compounds in the inhibitor libraries used for the screen should be given.

These were provided in the text (GSK via the SGC for PKIS1) and, for the “custom library”, in Table 2 (now Table 3) of the supplemental dataset; and this is now noted in the Methods.

Reviewer #3 (Remarks to the Author):

Review of “KiPIK, a new approach to identify kinases for specific phosphorylation sites”

In the manuscript, Dr. Higgins and coworkers developed a new technology to identify the specific kinase that phosphorylates any given phosphorylation site that they termed “KiPIK”. Using large-scale kinase inhibitor libraries paired with pre-existing *in vitro* kinase inhibitor response databases, they are able to generate inhibition profiles providing a “fingerprint” that allows unknown kinases that phosphorylate a specific phospho-site in cell lysates to be determined. Furthermore, they validate KiPIK’s effectiveness to identify unknown kinases in cell lysates using 2 phospho-sites with established kinases (Histone H3 and EGFR Y1068) and 1 phospho-site in INCENP with no known kinase. The authors then establish that the commonly phosphorylated motif in INCENP was a non-conical target of Cyclin B-Cdk1. Finally; the authors show that KiPIK has advantages over pre-existing methods for assigning kinases to phospho-sites such as RNA interference strategies by eliminating detection of indirect effects observed by inhibition of upstream kinases. As the current methods for assigning kinase phosphosite dependencies are limited, the ability to rapidly identify kinases using KiPIK would be tremendously advantageous, particularly in understanding kinase biology and development of resistance to kinase inhibitors.

Major concern:

Although the authors elegantly demonstrate that KiPIK can identify a previously unknown kinase that phosphorylates INCENP, further demonstration of the utility of KiPIK to identify unknown kinases will be required before the manuscript should be considered for publication. In order for the technology to be termed “generally applicable” at least 3 more examples of KiPIK need to be performed, including phosphorylation sites with no consensus motifs.

The reviewer does not explain his/her logic for requesting 3 more “unknown” examples. As mentioned earlier, we believe that to validate a new screening method, the most logical approach is to use kinase-substrate relationships that are already widely known and have stood the test of time, and we provide four of these. The problem with using new kinase-substrate pairs to validate the method is that this validation can only be as good as the limited number of follow-up assays that we can feasibly carry out on each one. Indeed, many published papers are largely occupied with fully validating a single new kinase-substrate relationship. To us, finding new unknowns therefore appears to be an inferior approach. Nevertheless, we agree that further examples can only enhance the paper and we have now added another instance of the use of KiPIK to discover an unknown kinase (for BCL9L), together with extensive post-KiPIK validation (please see response to reviewer 2, point 4). We are not completely certain what the reviewer means by “phosphorylation sites with no consensus motifs”. Presumably most kinases do recognise key features of their substrates, even if these consensus motifs have not yet been determined. We assume that the reviewer would like us to show that KiPIK can identify the correct kinase when this cannot be done from the sequence context of the phosphosite alone. We would argue that we have already shown that KiPIK outperforms consensus-based *in silico* approaches in this respect, as shown in Supplemental Figure 33. Prediction algorithms were unable to consistently identify the correct kinase for any of the 6 substrates we tested here. There was no consistent prediction for H3T3ph; PKA not Aurora B was the clear

best prediction for H3S28ph; EGFR was most often predicted for Integrin β 1Y795 rather than Src; and PKC was more commonly identified as the INCENP S446 kinase than Cdk1. The possible exceptions are EGFR Y1016 and BCL9L S915, where 4 of 6 methods identified EGFR and PKA respectively. Note that we exclude NetworkKIN here because it does not rely on sequence context alone, and even this algorithm did not rank the correct kinase as number one for 5 of the 6 substrates.

[redacted]

Taken together, the manuscript is well written and provides a potentially valuable technology for assigning unknown kinases for specific phosphorylation sites, however the manuscript is too preliminary in its current form.

REVIEWERS' COMMENTS:

Reviewer #2 (Remarks to the Author):

A method related to the one in this study has recently been published. See PMID: 31959955. This published technique also uses known information on the specificity of kinase inhibitors to discover kinases that phosphorylate peptide substrates, and it is therefore conceptually similar to KiPIK.

The authors should discuss how KiPIK differs from, and complements, the method in PMID: 31959955.

Response to Reviewers

Reviewer #2 (Remarks to the Author):

A method related to the one in this study has recently been published. See PMID: 31959955. This published technique also uses known information on the specificity of kinase inhibitors to discover kinases that phosphorylate peptide substrates, and it is therefore conceptually similar to KiPIK.

The authors should discuss how KiPIK differs from, and complements, the method in PMID: 31959955.

We thank the reviewer for highlighting this paper, and we now cite it and briefly explain how it is different in the Discussion, page 13.